



# Characterization of size-segregated particles turbulent flux and deposition velocity by eddy correlation method at an Arctic site

Antonio Donateo[1], Gianluca Pappaccogli[1,2], Daniela Famulari[3], Mauro Mazzola[4], Federico Scoto[1,2], Stefano Decesari[5]

[1] Institute of Atmospheric Sciences and Climate (ISAC), National Research Council (CNR), Lecce, 73100, Italy

[2] Joint Research Center - ENI-CNR Aldo Pontremoli, Lecce, 73100, Italy

[3] Institute of BioEconomy (IBE), National Research Council (CNR), Bologna, 40129, Italy

[4] Institute of Polar Sciences (ISP), National Research Council (CNR), Bologna, 40129, Italy

[5] Institute of Atmospheric Sciences and Climate (ISAC), National Research Council (CNR), Bologna, 40129, Italy

*Correspondence to*: Antonio Donateo (a.donateo@isac.cnr.it)

**Abstract.** Estimating aerosol depositions on snow and ice surfaces and assessing the aerosol lifecycle in the Arctic region is challenged by the scarce available measurement data for particle surface fluxes. This work aims at assessing the deposition velocity of atmospheric particles at an Arctic site (Ny-Ålesund, Svalbard Islands) over snow, during the melting season and over dry tundra. The measurements were performed using the eddy-covariance method from March to August 2021. The measurement system was based on a condensation particle counter (CPC) for ultrafine particle (UFP, < 0.25 μm) fluxes and an Optical Particle Counter (OPC) for evaluating particle size fluxes in the accumulation mode (ACC, 0.25 < dp < 0.7 μm) and quasi-coarse mode (CRS, 0.8 < dp < 3 μm). Turbulent fluxes in the ultrafine particle size range resulted prevalently downward especially in summertime. By contrast, particles fluxes in the accumulation and quasi-coarse mode were more frequently positive especially during the colder months, pointing to surface sources of particles from e.g., sea spray, snow sublimation or local pollution. The overall median deposition velocity ($V_d^+$) values were 0.90, 0.62 and 4.42 mm s$^{-1}$, for UFP, ACC and CRS, respectively. Deposition velocities were smaller, on average, over the snowpack with median values of 0.73, 0.42 and 3.50 mm s$^{-1}$. The observed velocities differ of less than 50% respect to previous literature in analogous environments (i.e. ice/snow) in the particles in the size range 0.01-1μm. At the same time, an agreement with the results of predictive models was found for only a few parameterizations, especially with Slinn (1982), while large biases were found with other models especially in the range 0.3 – 10 μm of particle diameters. Our observations show better fit with the models predicting a minimum deposition velocities for small accumulation mode particle sizes (0.1 - 0.3 μm) rather than for larger ones (~ 1 μm), which could result from an efficient interception of particles over snow surfaces which are rougher than the idealized ones. Finally, a polynomial fit parameterization was proposed in this study to describe the deposition velocity observations which properly represents their size-dependence and magnitude.

## 1 Introduction

The Arctic region is experiencing rapid climate change in response to the increase of greenhouse gases, aerosols, and other climate drivers and it is warming two to three times faster than the global average (Stjern et al., 2019), as



indicated both by observations and climate models (Cowtan and Way, 2014; Hartmann et al., 2013). Although this
phenomenon, known as Arctic amplification (Schmale et al., 2021), is mainly driven by changes in anthropogenic
greenhouse gases and climate feedbacks, short-lived climate forcers such as methane, tropospheric ozone, and aerosols
contribute to the observed environmental change (Arnold et al., 2016; Law et al., 2014; Quinn et al., 2008; Sand et
al., 2015; AMAP 2015a-b). The most important factors controlling aerosol climate forcing in the Arctic are: the long-
range transport of aerosols from mid-latitudes, local sources from both terrestrial and marine emissions, the surface
energy budget at high latitudes, the annual cycle of the cryosphere, and atmospheric depositions (Quinn et al., 2008;
Willis et al., 2018). Aerosol depositions on uncontaminated snow and ice can be a key factor affecting melting
processes (Skiles et al., 2018), because particles can decrease snow/ice albedo directly (as particles can contain light-
absorbing materials) or indirectly by affecting ice metamorphism. This phenomenon causes a reinforcing feedback
that melts the snow and ice, exposes darker underground leading to further surface warming (di Biagio, 2020, Abbatt
et al., 2019). The aerosols lifetime and hence the radiative forcing are strongly influenced by wet and dry deposition
processes (e.g., Johnson et al., 2018). Dry deposition is a complex process that is influenced by the microphysical
properties of aerosols and their sources, meteorological conditions, and surface morphological characteristics
(Donateo and Contini, 2014; Urgnani et al., 2022). The particles exchange between the atmosphere and the surface is
controlled also by frictional drag and terrain-induced flow modification (Giorgi, 1986; Stull, 1988). Knowing the
factors controlling dry deposition allows to estimate the residence time of particles in atmosphere, which governs their
transport distance and potential climate effects (Nemitz et al., 2002; Pryor et al., 2008). The accurate quantification of
particle deposition rates is a necessary prerequisite for modelling of aerosol cycles, particle size distribution, long-
range transport and radiative forcing potential (Wang et al., 2011; Liu et al., 2011; Zhou et al., 2012; Browse et al.,
2012; Menegoz et al., 2012; Lee et al., 2013; Eckhardt et al., 2015; Qi et al., 2017;). Despite its importance, however,
the size-dependence of deposition rates is poorly understood, and therefore the aerosols atmospheric lifetime remains
uncertain (Farmer et al., 2021; Emerson et al., 2020). Many aerosols properties have been investigated in the Arctic
with regards to chemical composition (Quinn et al., 2009; Köllner et al., 2021), total number and mass concentrations
(Croft et al., 2016), aerosol optical properties (Ferrero et al., 2019), ability to act as cloud condensation nuclei
(Bulatovic et al., 2021), and number and size distribution (Lupi et al., 2016; Song et al., 2022). However, relatively
few cases exist of aerosol deposition measurements on snow or iced surfaces, especially using the direct eddy-
correlation (EC) method (Farmer et al., 2021). These limitations are mainly due to the logistical challenges of
collecting continuous data sets in remote areas (Abbatt et al., 2019) and micrometeorology-based measurement
techniques, which require fast and sensitive detectors for EC (Burba et al., 2022). Dry deposition is typically described
by deposition velocity $V_d = -F/C$ , where F is a flux and C is concentration of the species of interest. $V_d$ provides a
particularly useful metric for comparing results across sites and for modelling particle removal because it is
independent of ambient concentration (Farmer et al., 2021). Further, the current understanding of Arctic Amplification
is limited by a lack of robust model representations of regional Arctic feedback processes with major challenges in
representing aerosol sources and sinks processes (Schmale et al., 2021). In particular, to reduce model uncertainty, a
deeper knowledge of the dry deposition velocity as a function of particles size is required and, more, an assessment
of current deposition models against an observational dataset of aerosol fluxes over the cryosphere is clearly needed



(Farmer et al., 2021). Global model skills in simulating the Arctic aerosol behaviour have improved in the last years (Eckhardt et al., 2015; Arnold et al., 2016), however, significant discrepancies remain in our understanding of Arctic aerosol deposition and removal (Saylor et al., 2019; Emerson et al., 2020; IPCC, 2021). Global models generally make use of an aerosol deposition module with a particle size–dependent resistance approach developed for specific
deposition surfaces (Slinn, 1982; Wesely and Hicks, 2000). The model developed by Zhang et al. (2001) expanded on the Slinn approach by incorporating simple empirical parameterizations for dry deposition processes. Zhang et al. (2001) also expanded the application of the resistance approach including also ice/snow surfaces. Deposition rates described by Zhang et al. (2001) have been compared to some observations over vegetated surfaces, however the parameters used to tune modern deposition models over cryosphere have not been tested against observations (Khan
and Perlinger, 2018).

The principal aim of this work is to measure the particle number fluxes and the related dry deposition velocities for size segregated particles (from ultrafine to quasi-coarse range) at an Arctic site located in the Svalbard Archipelago (Norway). Aiming to characterize the effect of surface properties on dry deposition, we performed continuous observations from the coldest months (on snow surface), to the snow melting period and all through the early summer
(snow-free surface). For these three conditions, a parametrization of the deposition velocity as a function of particle diameters will be provided. Finally, a functional parametrization of the deposition velocity on micrometeorological friction velocity has also been obtained.

## 2 Methodology

### 2.1 Measurement Site

Aerosol fluxes were measured at the Gruvebadet laboratory located southwest of the village of Ny-Ålesund (78°55'N, 11°56'E) in the archipelago of Svalbard (Norway). The measurement campaign started on 15[th] March 2021 and lasted until 15[th] August, for a total of 5 months. The site is characterized by small hills and depressions (height differences below 10 m) whereas the land-cover is characterized by dry tundra or bare soil (Magnani et al., 2022) during summer months, when the snowpack disappears. Ancillary meteorological measurements of air temperature and relative
humidity were collected at 5m a.g.l. at the "Amundsen-Nobile Climate Change Tower" (CCT) located about 1 km north west of the Gruvebadet laboratory (Mazzola et al., 2016).



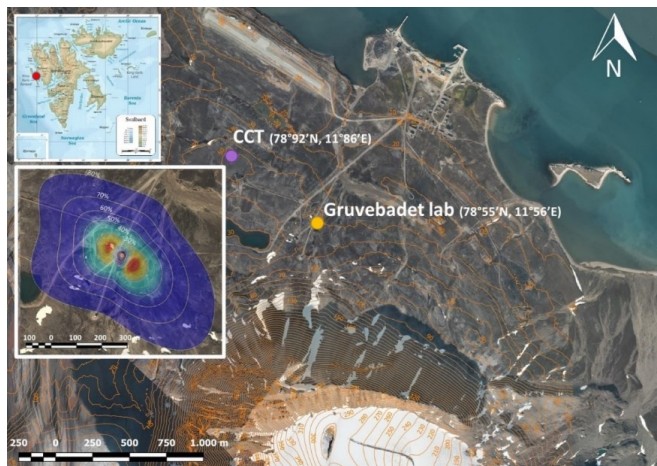

**Figure 1.** Location map of the study site: Ny-Ålesund (Svalbard, Norway). Purple and gold points indicate the Amundsen-Nobile Climate Change Tower and the Gruvebadet laboratory. Height contours (above sea level) are also presented. © Norwegian Polar Institute, www.npolar.no (accessed on 5/09/2022). In the inset (on the bottom left-hand corner) the flux footprint for the measurement setup is represented (see Section 2.4).

## 2.2 Instruments

The measurement system was located on the rooftop of the Gruvebadet laboratory at about 10 m (a.g.l.) on a pneumatic mast. Specifically, the EC station included an ultrasonic anemometer (Gill R3, Instruments Ltd., Lymington, UK) with a 100 Hz acquisition frequency, a condensation particle counter (CPC, TSI 3756) measuring the total particle number concentration and an optical particle counter (OPC, 11-D Grimm) measuring in 16 size channels (from 0.25 to 3 µm). In order to measure the particle concentration and fluxes from both CPC and OPC, a maximum acquisition rate of 1 Hz was used. Air was sampled by a unique inlet system for CPC and OPC. It was placed at a small distance from the sampling volume of the anemometer (about 25 cm). The air sample was driven into the laboratory by a silicon conductive tube 6 m long (internal diameter 26 mm), ending with a steel flow splitter (three-ways) 0.15 m long and 25 mm in diameter, in order to supply the air sample to both the CPC and the OPC. A nominal flow rate of 60 L min$^{-1}$ was applied to the flow splitter for the sampling operation giving a turbulent flow (Reynolds number equal to 4073) and minimizing the temporal distortion of concentration fluctuations. The CPC was connected to the flow splitter by a 0.82 m long silicon conductive tube (6 mm internal diameter) operating at a flow rate of 1.5 L min$^{-1}$, whereas a portion of 1.2 L min$^{-1}$ was aspirated by the OPC through a 0.80 m long (4 mm internal diameter) tube of the same type. The particle losses of the sampling system were calculated according to the formulation by Hinds (1999) for turbulent flow in the larger section tube (6 m) and according to Baron and Willeke (2001) for the laminar flow inside the two narrow tubes of the inlet system (beyond the flow splitter). The total particle losses amount to 13% (on average) for CPC and about 0.3% (on average) for OPC. According to TSI, the D50 of the TSI 3756 CPC is 2 nm under normal laboratory conditions. Particle penetration curve through the whole sampling system, calculated as the product of the penetration factors in the two tubing sections for CPC (Mordas et al., 2008; Kupc et al., 2013), shows that $D_{50}$ cut-off diameter (at 50% efficiency) is about 5 nm. Therefore, the system used was able to detect particles of between 5 and 1000 nm (the upper limit of the CPC). Both particle instruments were running together through a



specific software, developed by our research group, able to synchronize measured data with the anemometer output.
Air temperature and relative humidity were measured at CCT (5 m a.g.l.) by a conventional thermo-hygrometer
(Vaisala, mod. HMP45AC). Ten-minutes average physical size distributions above the canopy measured using a
Scanning Mobility Particle Sampler (SMPS) (TSI, mod. 3034) sampling from the base of the roof (5.5 m a.g.l.) were
corrected for particle size-specific tubing losses and then used to compute the particle number size distributions for
each half hour in the size range from 10 to 470 nm. More details of the size particle distribution measurements which
were conducted at Gruvebadet laboratory are given in Lupi et al. (2016). Precipitation data were measured during the
campaign by a laser-optical disdrometer Parsivel[2] (OTT, Messtechnik, Germany) installed on the roof of Gruvebadet.
The Parsivel can measure the size and fall speed of hydrometeors for a comprehensive measurement of all precipitation
types (rain, snow and hail). The Parsivel disdrometer can measure droplet sizes from 0.25 mm to about 25 mm, with
32 classes of varying diameter intervals. The velocity categories range from 0 m s$^{-1}$ to 22.4 m s$^{-1}$, with 32 classes of
varying intervals. Details of the instrument and the measurement technique used to determine the size and velocity of
hydrometeors can be found in the literature (e.g., Löffler-Mang and Joss, 2000; Battaglia et al., 2010; Tapiador et al.,
2010). An automatic snow meter station was located close to Gruvebadet laboratory at about 100 m SE during the
campaign period. It automatically provides continuous snow data, including near infrared images of the snow-cover
area, snow depth, internal snow temperature and liquid water content profiles at different depths with a time resolution
of 10 minutes.

### 2.3 Eddy covariance data analysis

The eddy covariance (EC) micrometeorological technique was applied in this work to quantify atmosphere-surface
particles exchange in size dependent mode. The flux is computed as the mean cross-product of the fluctuations of a
scalar concentration (s′) and the vertical component of wind velocity (w′), as $F_s = \overline{w's'}$, where the over-bar indicates
the average values (Kaimal and Finnigan, 1994; Stull, 1988). Turbulent fluxes were calculated on a 30-min basis using
a homemade code, developed in Matlab® 2018b. In this work, the micrometeorological convention was used,
according to which upward fluxes are positive, thus corresponding to emissions while downward fluxes (towards the
surface) are negative. EC measurements allow us to calculate the main turbulence characteristics such as the sensible
heat flux ($H = c_p\rho\overline{w'T_s'}$), where $T_s$ represents the sonic temperature, $c_p$=1005 J Kg$^{-1}$K$^{-1}$ the specific heat at constant
pressure, and $\rho$ the air density; the turbulent kinetic energy $TKE = 1/2(\sigma_u^2 + \sigma_v^2 + \sigma_w^2)$, where $\sigma_{u,v,w}$ is the standard
deviation of the wind components; the friction velocity $u_* = \left(\overline{u'w'}^2 + \overline{v'w'}^2\right)^{1/4}$, where $\overline{u'w'}$ and $\overline{v'w'}$ are the
horizontal momentum fluxes. From CPC and OPC measurements together, a complete characterization of particle
number concentration can be obtained, from ultrafine to coarse particle size ranges on a whole of 17 size bins. Particle
fluxes from particle number concentration were calculated according to $F_{Ni} = \overline{w'ci'}$ for each particle size bin (index
$i$). Size-resolved deposition velocities ($V_{di}$) of aerosol particles were defined according to

$$V_{di} = - F_{Ni}/N_i \tag{1}$$

namely the turbulent flux of each stage normalized by the respective particle number concentration. The minus sign
is to define positive values of deposition velocity ($V_d^+$) as a transport toward the surface (deposition) and negative





values ($V_d^-$) as a transport into the atmosphere (emission), respectively. Positive and negative fluxes will be treated as two separate processes following uneven distribution of positive and negative fluxes around zero. Separation of upward and downward flux periods or systematic removal periods of positive flux based on other parameters is a common practice in deposition studies (e.g. Nilsson and Rannik, 2001; Vong et al., 2004; Pryor et al., 2013; Lavi et al., 2013; Emerson et al., 2020). Atmospheric stability is a crucial quantity in characterizing turbulent fluxes pollutant

dispersion in the environment, and it is considered in parametrizations of turbulent characteristics and wind profiles (Nordbo et al., 2013). In order to classify the different atmospheric conditions over all measurement period, the atmospheric stability is defined as $\zeta = z/L$, where $z$ is the measurement height and $L$ represents the Obukhov length, computed as:

$$L = \frac{-u_*^3 \overline{T}}{\kappa g \overline{w'T_s'}}$$ (2)

where $T_s$ is the sonic temperature measured by the anemometer. Five stability classes are estimated according to Nordbo et al., (2013) as follows: very unstable ($\zeta < - 1$) and unstable ($\zeta < - 0.01$), neutral ($- 0.01 \leq \zeta \leq 0.01$), stable ($\zeta > 0.01$) and very stable ($\zeta > 1$).

**2.4 Data pre-processing**

Half-hours hard-flagged for drop-outs, discontinuities for power loss or inputs outside the absolute limits were

discarded from the dataset. These events, including also a quality control process, resulted in 25% and 24% of the data being rejected for the CPC and OPC fluxes, respectively. OPC measurements ended on 29[th] of July for instrumental technical issues. Raw time series were pre-processed applying, first of all, a despiking procedure to eliminate spurious spikes due, for example, to electronics issues. Spikes in the 100 Hz (anemometer) and 1 Hz (CPC and OPC) time series were removed from the dataset and replaced by linear interpolation using a procedure described by Vickers and

Mahrt (1997). A spike detection algorithm was applied on raw high-frequency data which defines spikes as absolute deviations from the mean of a threshold value, that is 6-fold σ (where σ is the variance of 10 min sub-interval). Using a closed path instrument (i.e. CPC or OPC) with a sampling tube, air sampled by the sonic anemometer gets to the scalar sensor many seconds later than the vertical wind signal. Without correcting for the time delay, vertical wind component fluctuations $w'$ do not correlate properly with concentration fluctuations $c'$, leading to wrong flux

estimation. Such time lag was estimated by means of cross-correlation analysis by moving time series forward, defining the maximum covariance between the vertical component of wind speed and particle number concentration, considering a time window between 3 s and 5 s, which was estimated through the flow rate, tube length and response time (Deventer et al., 2015). A mean time lag of 4.14 s and 3.77 s was observed, respectively for CPC and OPC (for all size channels) measurements. Data were rotated in the streamline reference system (McMillen, 1988) with three

reference rotation. The first two rotations are always performed and they select a reference system that, for each averaging period, align the streamline velocity component with the direction of the mean velocity vector. The third rotation was not performed only if the absolute value of the angle of attack (Nakai and Shimoyama, 2012) is greater than 15° (about 3% of total cases). Critical conditions for the applicability of the Monin Obukhov similarity theory occur with neutral or stable atmospheric conditions, low wind speed, weak and intermittent turbulence (Sun et al.





2012, Schiavon et al., 2019), and significant effect of sub-meso motions (Vickers and Mahrt 2006; Liang et al., 2014). The former cause, for example, the meandering of the velocity vector (Mortarini et al., 2016) and do not follow surface-layer similarity, although, they can contribute to observed statistics, especially when small-scale turbulence is weak. According to some authors (e.g., Vickers and Mahrt 2003), similarity relationships should be evaluated only after filtering out the contribution from these motions. In this work, the energy contributions related to non-turbulent,

sub-meso motions, with time scales often longer than the investigated time window, were removed by a recursive digital filter both for heat and particles fluxes (Falocchi et al., 2018; Pappaccogli et al., 2022). The recursive digital filter worked on a different time scale, according to atmospheric stability conditions. Ogive analysis (Fig. 2b) was carried out in order to estimate a properly time scale (Metzger and Holmes, 2008). A detailed description of the methodology used for spectral and ogive analysis is reported in Pappaccogli et al. (2022). Time scale for unstable

atmosphere is 522 s, whereas the value decrease to 350 s and 340 s for neutral and stable conditions, respectively. This filter does not introduce any phase shift or signal amplitude attenuation in the filtered time series. The filtering procedure used two data buffer (1800 s long), before and after the considered 30 min period of investigation. A fundamental assumption of the EC method is that fluctuations are statistically stationary during the chosen averaging time in order to ensure the calculation of an ensemble average. A stationarity tests as reported by Mahrt (1998) were

carried out on sonic temperature and particles concentration fluctuations (Cava et al., 2014; Věcenaj and De Wekker, 2015). A lower detection limit for the fluxes in the sampling system was computed using the method proposed by Langford et al. (2015) and defined at 2.8 $cm^{-2}$ $s^{-1}$ for the CPC and 0.3 $cm^{-2}$ $s^{-1}$ for the OPC. In order to ensure exclusively the study of particle dry deposition, all data corresponding to a precipitation intensity greater than 0.1 mm $h^{-1}$ for a time period greater than 5 min (on the averaging period of 30 min) were also rejected. Error associated to the

random and limited statistical counting error (relative %) in the particle number measurements was estimated through the approach reported in Deventer et al. (2015) for number concentration $\delta(N)$ and fluxes $\delta(w'N')$, while the method reported in Fairall (1984) was used for the deposition velocity $\delta(V_d)$ for each size range (Table 1). Obviously, uncertainties due to discrete counting $\delta(N)$ are negligibly small for all stages, by the way increasing from ultrafine to coarse particles (greater than 100%). Hence, also the relative flux uncertainty $\delta(w'N')$ due to limited counting statistics

is moderate (on average 9%), and the average exceeds 25% only for the 11[th] OPC channel. If the counting errors on deposition velocity $\delta(V_d)$ is considered, on first size channel (CPC) it was very low (< 5%). The same error for the first eleven channels of OPC (0.25 μm - 0.80 μm) was on average about 85%, while for the remaining channels (1 μm - 3 μm) it was on average greater than 100% (on average 109%). In order to lower the associated statistical counting error, especially on deposition velocity, the first nine channels have been pooled together as the rest of the seven

channels (Whitehead et al., 2012; Conte et al., 2018; Donateo et al., 2019).

**Table 1.** Sampling system specifications with aerodynamic ($D_{50}$) cut points and the respective geometric mean diameter ($D_{gm}$) as well as mean statistics of the signal quality evaluation.


|  | $D_{50}$ (μm) | $D_{gm}$ (μm) | δ (N) % | δ (w'N') % | δ ($V_d$) % |
|---|---|---|---|---|---|
| UFP | 0.005 | 0.035 | 4.52E-06 | 0.22 | 3.9 |
|  | 0.25 | 0.26 | 0.001 | 4.78 | 61 |



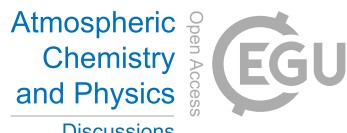

| | | | | |
|---|---|---|---|---|
| | 0.28 | 0.29 | 0.002 | 5.86 | 68 |
| | 0.3 | 0.32 | 0.004 | 7.38 | 74 |
| | 0.35 | 0.37 | 0.009 | 7.61 | 80 |
| ACC | 0.4 | 0.42 | 0.025 | 5.02 | 84 |
| | 0.45 | 0.47 | 0.065 | 8.59 | 90 |
| | 0.5 | 0.54 | 0.14 | 6.39 | 93 |
| | 0.58 | 0.61 | 0.27 | 5.71 | 94 |
| | 0.65 | 0.67 | 0.45 | 6.30 | 95 |
| | 0.7 | 0.75 | 0.72 | 12.74 | 98 |
| | 0.8 | 0.89 | 1.24 | 26.67 | 99 |
| | 1 | 1.14 | 2.37 | 6.07 | 102 |
| CRS | 1.3 | 1.44 | 5.38 | 9.06 | 106 |
| | 1.6 | 1.79 | 11.47 | 12.62 | 109 |
| | 2 | 2.24 | 84.49 | 6.54 | 112 |
| | 2.5 | 2.74 | >100 | 11.98 | 117 |
| | 3 | | | | |

**Table 2.** Sampling system specifications with aerodynamic ($D_{50}$) cut points and the respective geometric mean diameter ($D_{gm}$) as well as mean statistics of the signal quality evaluation.


| | $D_{50}$ (µm) | $D_{gm}$ (µm) | δ (N) % | δ (w'N') % | δ ($V_d$) % |
|---|---|---|---|---|---|
| UFP | 0.005 | 0.035 | 4.52E-06 | 0.22 | 3.9 |
| ACC | 0.25 | 0.42 | 1.84E-4 | 0.63 | 29 |
| CRS | 0.8 | 1.45 | 0.0167 | 0.63 | 41 |
| | 3 | | | | |

The previous considerations confirm that it is possible to aggregate the particles in the three mentioned size ranges that are indicated in the following as ultrafine (UFP, 5 nm < $d_p$ < 0.25 µm), accumulation (ACC, 0.25 < $d_p$ < 0.7 µm), and quasi-coarse (CRS, 0.8 < $d_p$ < 3 µm) mode, the last indicating a particle size range between large accumulation

mode and small coarse particles. It is worth to remark that UFP particle concentration has been obtained as difference between the total number concentration (CPC measurement) and the OPC total concentration in the size range 0.25-1µm. The relative counting errors on $V_d$ for these two groups are about 29% for ACC and 41% for CRS. This size aggregation is also confirmed by the correlation analysis of the concentration time series associated with the different size classes. The resulting correlation shows that all classes in the first and second group have a good temporal

correlation (Pearson coefficient > 0.6) with each other. UFP fraction was obtained from the difference between the sum of the first 12 OPC channels (up to 1 µm) and the total number concentration of the CPC. This subdivision will be useful for studying the characteristics and trends of the particles and their possible correlation with meteorological and micrometeorological parameters.

**2.5 Spectral analysis and corrections**



The cospectra were computed on the vertical wind speed component and the analysed scalars (sonic temperature, ultrafine particle concentration and accumulation and coarse particle concentration) by means of the Fast Fourier Transform (FFT). Figure 2a shows normalized median cospectra (about 500) of kinematic heat flux, ultrafine particle number flux and accumulation-coarse particle number flux as a function of a normalized frequency $f_n = fz/u$, where $u$ is the mean wind velocity). The cospectra were all calculated over one-hour periods. ACC-QCRS cospectra were

calculated based on the sum of all OPC size channels. The cospectra $Co_{w'x'}$, between vertical wind velocity and the scalar $x$ are normalized with the correlation $\overline{w'x'}$. Cospectra calculated for all variables display similar patterns, showing a -7/3 decay for frequencies above 1 Hz, within the universal equilibrium range.

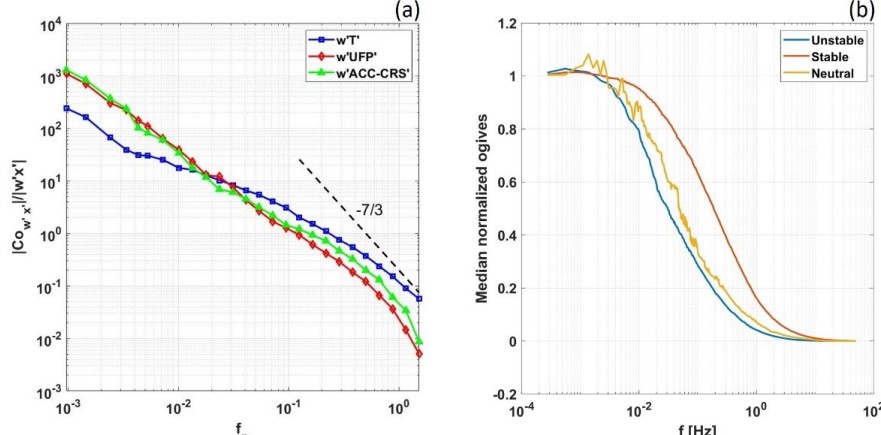

**Figure 2**. (a) Normalized median cospectra of kinematic heat flux (blue), ultrafine particle concentration flux (red) and accumulation-coarse particle concentration flux (green) as a function of $f_n$. The binned median cospectra (about 500) were computed from continuous period of 1h. Black dashed line represents the slope theoretically predicted in the inertial subrange (Kolmogorov, 1941); (b) median normalized ogives for kinematic heat flux as a function of natural frequency (Hz) for the three classes of atmospheric stability.

Specifically, kinematic heat flux cospectrum shows a slight slope at the high frequencies whereas tend to level-off in the low-frequency range. Results indicate that the contribution to the correlation between w and particle concentration follow kinematic heat flux trend up to normalized frequencies in the interval 0.5-0.8. Therefore, the system used was generally able to measure concentration fluctuations at frequencies that made a substantial contribution to the vertical turbulent fluxes of particles. Figure 2b shows the normalized median ogives of kinematic heat flux. Comparison of

the ogives for the different atmospheric stability classes shows that the higher frequencies contribution is larger under stable conditions. Indeed, no relevant contribution to kinematic heat flux occurs above 1 Hz (1 s time scale) under unstable conditions, whereas a small turbulence contribution is still present up to about 2 Hz (0.5 s) and 5 Hz (0.2 s) under neutral and stable conditions respectively. Ogives obtained for neutral condition display some irregularities due to the low number of available cases. Spectral corrections were applied to the UFP, ACC and CRS particle number

fluxes using the theoretical approach described in Aubinet et al. (2012). The first-order time constant of the CPC and OPC measurement systems were determined by estimating the time response (at first order) to a concentration step in a laboratory experiment. The results were $\tau_{CPC}= 0.77$ +/- 0.01 s and $\tau_{OPC} = 0.40$ +/- 0.03 s (identical for each size





channel). High frequency under sampling of the EC system, due to the relative low frequency response of the CPC and OPC, leads to an error (underestimation) on particles fluxes measurement. It means that the full atmospheric

cospectra between the vertical wind velocity and particle concentration can be under sampled at high frequency (Fig. 2a). High frequency losses were corrected following the approach developed by Horst (1997) and they have been quantified in 30% for CPC and 21% for OPC. Spectral correction for the attenuation of the fluctuations during the transport of the aerosol sample in the inlet tube was obtained according to the method reported by Massman and Ibrom (2008). This spectral correction for our measurement system resulted in a very low associated error (lower than

285    0.01%).

### 3 Results and Discussion

### 3.1 Site Meteorology and Micrometeorology

During the measurement period the air temperature was about -10°C in March and April, while during summertime (i.e. June-August) the temperature was above 5°C (Fig. 3a).

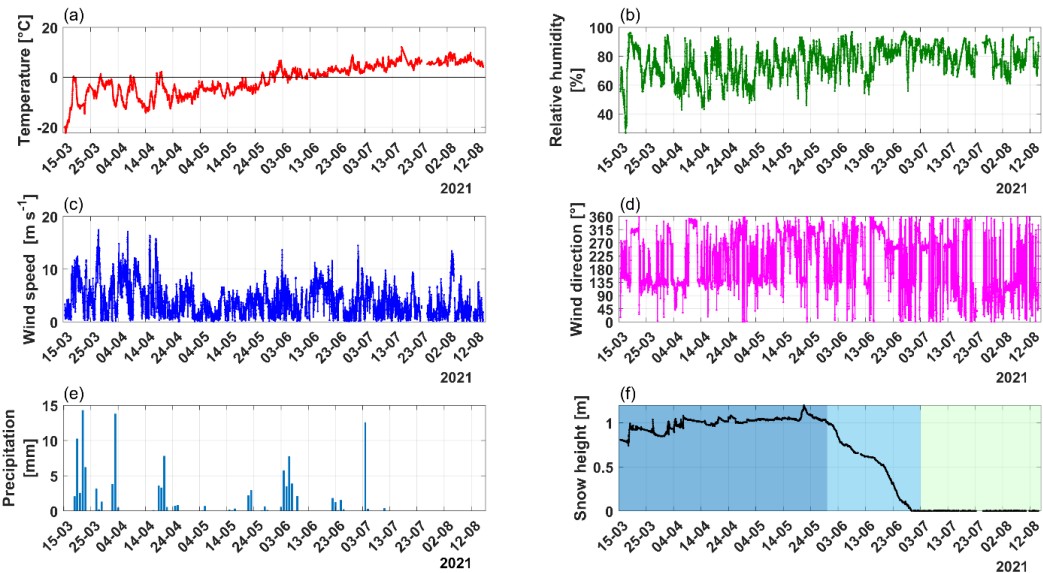


**Figure 3.** Time series of the principal meteorological variables measured during the campaign in Ny-Ålesund. (a) air temperature (°C), (b) relative humidity (%), (c) wind velocity (m/s), (d) wind direction (degree N), (e) daily cumulated precipitation (mm), and (f) snow depth (m). In (f) panel three colour bands were used to separate the period characterized by snow cover (dark cyan) from melting snow period (cyan) and, finally, snow-free phase (light cyan).


Positive temperature values occur approximately at the end of May (29th May), leading to a gradual melting of the snowpack, which reached a maximum height of 1.20 m on 21th May (Fig. 3f). Specifically, the snowpack had an average height of 1 m until 30th May, hereafter named snow cover period (SC). Snow depth decreased gradually for





about 8 days due to its compacting processes. Snowpack was on average 0.54 m from 30[th] May for about a month up

to 3[rd] of July. In this period (hereafter snow melting period, SM), snowpack went through a complete melting until

the summer period with the presence of widespread ponds and snow patches in the footprint of the EC system. This

period was characterised by positive daily average temperatures. The last period of the measurement campaign, from

3[rd] of July to 15[th] August will be referred to as snow-free period (SF), with the surface covered by dry tundra. Relative

humidity in general increases from a minimum of 37% during a severe storm in the month of March, up to 82% in

July (Fig. 3b).

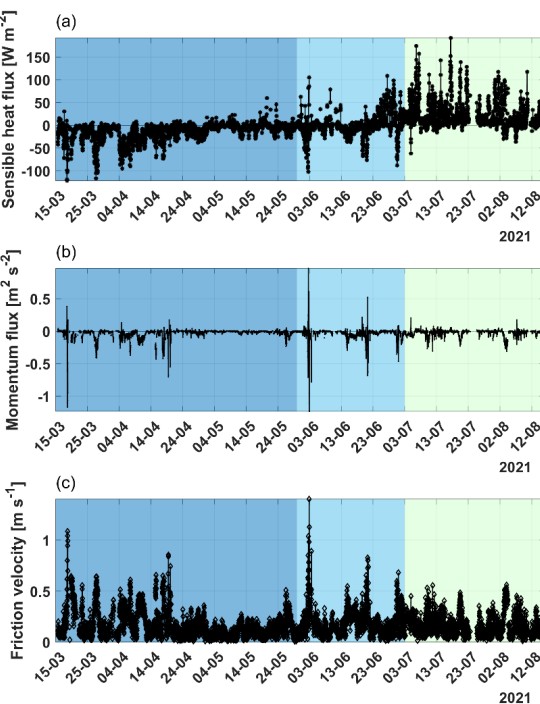

**Figure 4.** Time series of the principal micrometeorological variables measured during the campaign carried in Ny-Ålesund. (a) sensible heat flux, (b) momentum turbulent flux, and (c) friction velocity u*. In each panel three colour bands were used to separate the period characterized by snow cover (dark cyan) from melting snow period (cyan) and, finally, snow-free phase (light cyan).


A direct correlation is observed between snow melting and increased relative humidity, even if the latter variable is

also strongly influenced by the presence of the Kongsfjorden close to the site. Two prevailing wind directions can be

distinguished: one characterised by winds coming from the NW sector for 27 % of the cases and one with winds

coming from the SE for 42 % of the cases (Fig. 3d). Indeed, the prevailing wind directions are essentially along the

direction of the Kongsfjorden. On the other hand, at mid-end June onwards, wind field shows two components from

W-SW and NE (with low winds). The highest wind speeds are measured in the NW direction with an average of 4.9

m s[-1], while in the SE direction the average wind speed is about 4.0 m s[-1] (Fig. 3c). In the measurement period 22 rain

days were observed starting from 1[st] June, with a cumulative precipitation above 2 mm for 9 days. In this period the



maximum amount of cumulative rain on daily basis was of 23 mm on 4th June. In the first part of the campaign, until 30th May, 16 days of precipitation were observed, with a peak of snow height (1.20 m) on 20th May.

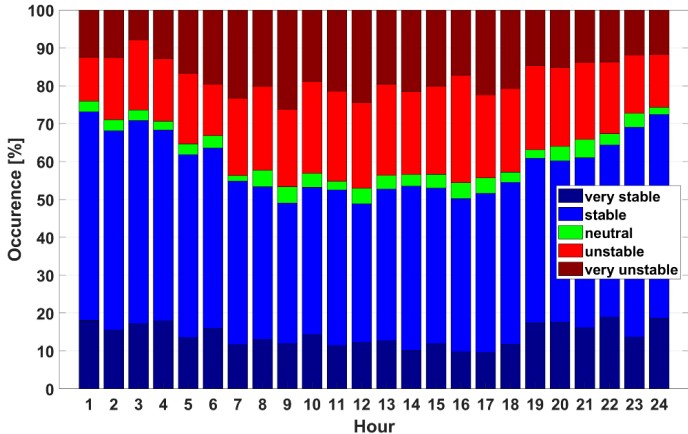

**Figure 5.** Percentage of atmospheric stability over the daily cycle for the investigated dataset. Stability classes based on threshold values of the stability parameter $\zeta$ were defined as: very unstable (dark red, $\zeta < -1$) and unstable (red, $\zeta < -0.01$), neutral (green, $-0.01 \leq \zeta \leq 0.01$), stable (blue, $\zeta > 0.01$) and very stable (dark blue, $\zeta > 1$).

The results from this work are completely in line with those reported in (Mazzola et al., 2016) for the data collected at CCT from 2010 to 2016. In the measurement period, sensible heat flux was on average negative (-0.8 W m$^{-2}$) (Fig. 4a). The results show the presence of a long period with negative energy fluxes (with an average value of -10.7 W m$^{-2}$) and a minimum around -121 W m$^{-2}$ while snow cover is present and during the snow melting phase, when the atmosphere is warmer than the surface. As the snow melts, positive sensible heat fluxes begun to appear, with values of up to 191 W m$^{-2}$ (on half hourly base) in the month of July. This behaviour has also been observed in other measurement campaigns in the Antarctica (Van den Broeke, 2005; Van As et al., 2005) and in the Arctic (Kral et al., 2014) where the snowpack acts as a sink of heat during the winter and spring months. The kinematic momentum flux was typically downward (Fig. 4b) with an average of -0.034 m$^2$ s$^{-2}$; however, for about 16% of the data, under conditions of wind direction from S-SW sector and average wind velocity of 2 m s$^{-1}$, the direction of the momentum flux changed, most likely due to the effect of swell (Grachev and Fairall, 2001). Figure 4c reports the time series of friction velocity, with a mean value of 0.17 m s$^{-1}$ over the whole measurement period. No specific differences can be noted in the friction velocity behaviour due to the changing in snowpack characteristics. As it can be noted in Fig. 5, stable conditions prevailed during the campaign, with 44% of frequency (of which 14% of cases very stable), while unstable stratification was less frequent (20% and 18% for unstable and very unstable, respectively). Neutral stratification occurred on 4% of the time, with no diurnal variability. It is worth noting the difference from a typical daily pattern at mid-latitudes, characterised by a much greater contrast between day and night PBL temperature profile structures, with more unstable conditions during the day and more frequent stable conditions at night.





### 3.2 Footprint analysis

The ratio between wind velocity and friction velocity in atmospheric neutral conditions was used to evaluate the average roughness length $z_0$, for the measurement site using a parameterisation based on similarity theory (Toda and Sugita, 2003). The results gave $z_0 = 0.08$ m if the whole measurement period is considered. Separating the period with snow coverage from that without snow, the calculated values were $z_0 = 0.19$ m and $z_0 = 0.43$ m, respectively. No

significant statistical differences were found exploring the roughness length for different wind sectors. Besides, evaluation by the method reported in Toda and Sugita (2003) gave a null displacement height, considering again all wind directions together. Source area for scalar fluxes have been evaluated using a Lagrangian footprint model by Kljun et al. (2015). Results of flux footprint analysis are shown in Fig. 1 (inset bottom left-hand corner) with different influence level of the zones on the measurements. The particle fluxes measured at the rooftop site represented a surface

area of about 0.48 km² (with respect to the 80% contour line) with a maximum distance of 400 m both in north-west and south-east direction (otherwise 300 m in NE and SW sector). The flux peak contribution was in the wind direction sectors at a distance of about 40 m ($\pm$ 20 m) (Fig. 1). However, the source areas were quite homogeneous for all wind direction sectors around the measurement site, with 100% of snow coverage for the first period of the campaign; from then onwards, the land cover was largely dry tundra and bedrocks, with about 1% water surfaces (an arctic lake) in

the flux footprint (Fig. 1).

### 3.3 Particle Concentration and Turbulent Fluxes

Average particle number concentration, over the whole measurement period, for UFP (hereafter $N_{UFP}$) was 595 cm⁻³ (median 238 cm⁻³), for ACC mode (hereafter $N_{ACC}$) was 25 cm⁻³ (median 19 cm⁻³), and for CRS mode (hereafter $N_{CRS}$) was 0.47 cm⁻³ (median 0.30 cm⁻³) (Table 3). $N_{UFP}$ increases from March (on average 166 cm⁻³) to August 2021,

reaching its maximum in July (951 cm⁻³) (Fig. 6a). On the other hand, $N_{ACC}$ and $N_{CRS}$ show a different, anti-correlated behaviour respect to the $N_{UFP}$ (Figs. 6b and 6c). $N_{ACC}$ and $N_{CRS}$ reach the maximum concentration in March with respectively 40 cm⁻³ and 0.67 cm⁻³, and then they decrease down to a minimum concentration in June (18 cm⁻³ for $N_{ACC}$ and 0.47 cm⁻³ for $N_{CRS}$). The same behaviour was observed by Croft et al. (2016) at the Mt. Zeppelin observatory. The complete annual cycle is remarkably similar to that observed at Mt. Zeppelin over an earlier 10-year period from

2000 to 2010 (Tunved et al., 2013). The particles size distribution was continuously monitored, combining the SMPS and OPC measurements, from the nanoparticle (4 nm) up to the micro particles (2.5 μm) size range. The total size distribution plot (Fig. 7 - black markers) referred to the whole measurement period. This plot shows a two-mode size distribution: with high concentration of particles in the 0.02 - 0.04 μm range with a peak at 0.03 μm (ultrafine mode) and 0.1 - 0.2 μm range with a peak at 0.15 μm (accumulation mode). Considering the snow cover period, particle size

distribution (red markers) also presented two modes: one peaked at 0.03 µm and the other at 0.15 µm; with respect to

the whole period analysis the dominant mode was the accumulation one.

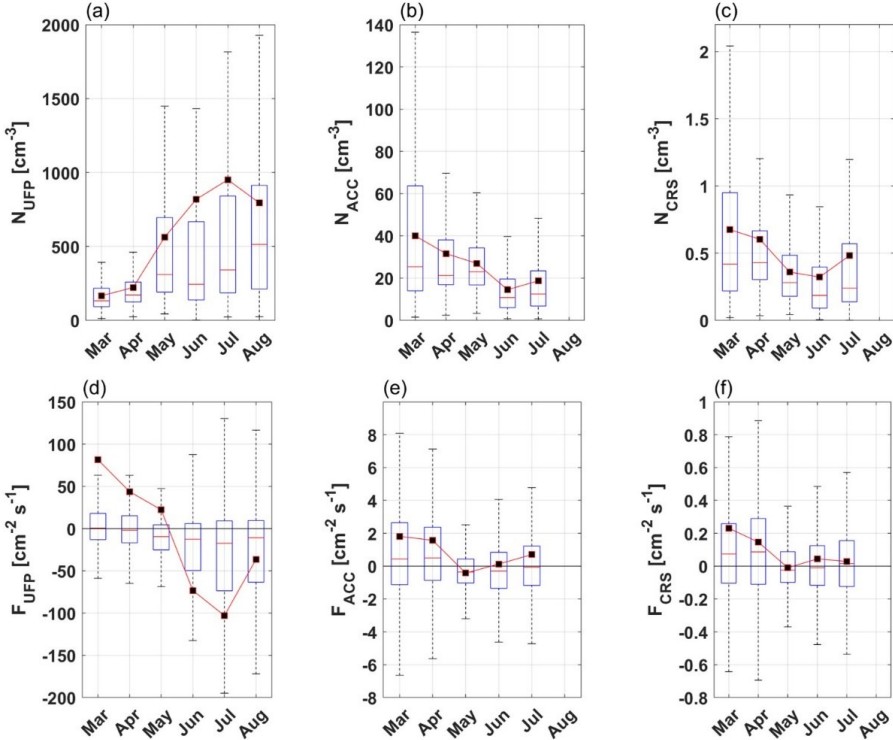

**Figure 6.** Upper panels, monthly box plots for a) $N_{UFP}$, b) $N_{ACC}$ and c) $N_{CRS}$. Bottom panels, monthly box plots for d) $F_{UFP}$, e) $F_{ACC}$ and f) $F_{CRS}$. Continuous red lines represent median values. value. Boxes represent the 25th and 75th
percentile. Whiskers correspond to ± 2.7σ and 99.3% data coverage. In black (square) are the average values for the measured variables.

Proceeding towards the melting period and all through the snow-free period, the two-modes size distribution tends to

peak in the ultrafine range (0.02-0.03 µm), with an increasing concentration. In particular, during the snow melting,

particle concentration in the accumulation mode decreased, while UFP mode significantly increases in this period.

Previous studies at the mountain station of Zeppelin observatory (south of Ny-Ålesund) showed that new particle

formation events seem to be a rather common phenomenon during the summer season, and this is the result of both

the photochemical production of nucleating/condensing species and low condensation sink (Tunved et al., 2013;

Ferrero et al., 2016). The particle size distribution pattern from the snow cover to the snow-free period can be explained

as follow: during March–May, there is sunlight but also a large deposition aerosol surface due to the presence of long-

range transported aerosol (Giardi et al., 2016). On the other hand, during June-August, atmospheric conditions are

quite different with high solar insolation and more daylight hours, bringing to new particles production and their

growth into a large size (accumulation mode). Further, seasonal effect of nucleation precursors is also related to





sources: marine biota (the 'phytoplankton blooms') grow in late spring and early summer emitting methane sulphonic
acid (MSA), which gives rise to nucleation (Beck et al., 2021).

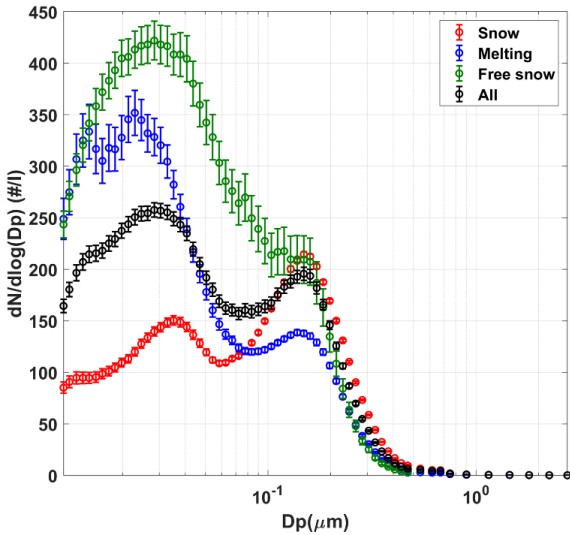

**Figure 7.** Particle size distribution for the measurement period

The turbulent fluxes of ultrafine ($F_{UFP}$), accumulation ($F_{ACC}$) and quasi-coarse ($F_{CRS}$) particles statistics are listed in
Table 3 considering the snow cover, melting and -free periods. The average $F_{UFP}$ was -16.99 cm$^{-2}$ s$^{-1}$ (median -8.20
cm$^{-2}$ s$^{-1}$) over the entire measurement period, specifically, the measurement site behaved, on average, as a deposition
area (negative fluxes) for particles in that size range both with and without snowpack (Table 3). $F_{ACC}$ for the whole
period was positive with an average value of 0.64 cm$^{-2}$ s$^{-1}$ (median -0.15 cm$^{-2}$ s$^{-1}$), but from snow melting onwards the
fluxes became negative in median values, showing a sink behaviour for the measurement site for this size range
particles. Finally, $F_{CRS}$ was 0.07 cm$^{-2}$ s$^{-1}$ (median 0.03 cm$^{-2}$ s$^{-1}$) on the whole period. Overall, in median values, quasi-
coarse mode fluxes are positive for all periods, even if, on average, these fluxes were very small. However, this net
deposition is not a steady feature. In fact, total and size-resolved turbulent fluxes are highly variable both in magnitude
and direction (emission and deposition). Turbulent fluxes in ultrafine mode are negative for 63% of the quality-assured
cases (Fig. 6d), establishing a deposition behaviour in this size range for the measurement site. Similarly, $F_{ACC}$ are
negative in 52% of validated cases, while $F_{CRS}$ is negative for 46% of available good flux cases. The values of $F_{UFP}$
are on median positive (0.59 cm$^{-2}$ s$^{-1}$) at the beginning of the measurement period, when ultrafine particle concentration
is small, then turn significantly negative (median -17.3 cm$^{-2}$ s$^{-1}$) in the summer when concentrations sharply increase
(Fig. 6a). If the high summertime concentrations of UFP are due to new-particle formation processes, these do not
occur near surface and propagate upward, but rather take place in the full mixing layer, topping at ≤ 500 m above the
ground (Ferrero et al., 2016) with UFP mixing down and deposited to the surface. $F_{ACC}$ and $F_{CRS}$ have generally a very
similar monthly behaviour (Figs. 6e and 6f) with a maximum value in the month of April (median 0.50 cm$^{-2}$ s$^{-1}$ and



0.09 cm$^{-2}$ s$^{-1}$, respectively) and a minimum in the month of May (median -0.35 cm$^{-2}$ s$^{-1}$ and -0.02 cm$^{-2}$ s$^{-1}$, respectively).
Although remote Arctic environments are normally considered receptors for pollution transported from mid-latitudes
and the seasonal cycle of atmospheric aerosol concentrations witnesses it, at surface level small deposition fluxes can

overlap to – and being exceeded by – small emission fluxes of particles. In Ny-Ålesund, beside an unquantified
contribution from the village itself and from shipping (Eckhardt et al, 2013), emission fluxes of natural aerosols can
originate from soil erosion (in the summer) as well as snow resuspension and sublimation. Finally, it is unclear whether
a residual sea spray emission can lead to detectable emission fluxes of sea salt particles at ~1 km inland from the coast.
Our measurements – relying on fast particle counting – do not provide direct insights to the chemical nature, properties

and actual origin of the aerosols associated with either emissions or depositions.

**Table 3.** Statistical quantities of UFP, ACC and CRS for particle number concentration and their turbulent fluxes for
the different phases considered in this work: snow cover, snow melting and snow-free period.

| | $N_{UFP}$ (cm$^{-3}$) | | | $N_{ACC}$ (cm$^{-3}$) | | | $N_{CRS}$ (cm$^{-3}$) | | |
|---|---|---|---|---|---|---|---|---|---|
| | SC | SM | SF | SC | SM | SF | SC | SM | SF |
| 10$^{th}$ prc | 80.25 | 56.59 | 106.87 | 10.89 | 3.98 | 4.03 | 0.13 | 0.05 | 0.08 |
| mean | 347.09 | 753.76 | 934.39 | 31.21 | 16.20 | 19.32 | 0.52 | 0.33 | 0.50 |
| std.dev | 509.08 | 1579.9 | 1421.7 | 24.71 | 13.99 | 18.10 | 0.54 | 0.38 | 0.68 |
| std.err | 8.51 | 40.03 | 32.80 | 0.41 | 0.35 | 0.54 | 0.01 | 0.01 | 0.02 |
| median | 197.20 | 247.42 | 410.51 | 22.13 | 11.92 | 12.39 | 0.36 | 0.20 | 0.24 |
| 90$^{th}$ prc | 750.45 | 1554.1 | 2572.4 | 66.82 | 34.59 | 44.66 | 1.06 | 0.75 | 1.12 |

| | $F_{UFP}$ (cm$^{-2}$ s$^{-1}$) | | | $F_{ACC}$ (cm$^{-2}$ s$^{-1}$) | | | $F_{CRS}$ (cm$^{-2}$ s$^{-1}$) | | |
|---|---|---|---|---|---|---|---|---|---|
| | SC | SM | SF | SC | SM | SF | SC | SM | SF |
| 10$^{th}$ prc | -49.21 | -174.0 | -319.0 | -2.44 | -3.10 | -2.53 | -0.25 | -0.30 | -0.28 |
| mean | 43.80 | -65.76 | -85.86 | 0.86 | 0.04 | 0.81 | 0.10 | 0.03 | 0.04 |
| std.dev | 1230.2 | 442.0 | 597.3 | 6.94 | 4.86 | 5.61 | 0.96 | 0.53 | 0.39 |
| std.err | 23.53 | 12.62 | 15.20 | 0.13 | 0.14 | 0.18 | 0.02 | 0.02 | 0.01 |
| median | -4.58 | -11.38 | -15.01 | 0.11 | -0.32 | -0.08 | 0.04 | -0.01 | 0.02 |
| 90$^{th}$ prc | 52.46 | 30.42 | 60.61 | 4.24 | 2.79 | 3.83 | 0.45 | 0.36 | 0.40 |


### 3.4 Particle deposition velocity $V_d$

Exchange velocity ($V_{ex}$) calculated from normalization of the turbulent fluxes (Eq. 1) on the number concentration, in
general, as seen in the previous Section 3.3, presents both positive and negative values, representing emission and
deposition events. $V_{ex}$ calculated for the measurement campaign resulted on median value 0.34 mm s$^{-1}$ (IQR$_{10-90}$ -2.00
to 2.81 mm s$^{-1}$) for ultrafine size range, 0.10 mm s$^{-1}$ (IQR$_{10-90}$ -2.02 to 1.53 mm s$^{-1}$) for the accumulation mode and -

1.36 mm s$^{-1}$ (IQR$_{10-90}$ -11.25 to 8.48 mm s$^{-1}$) for coarse range (Table 4). IQR$_{10-90}$ represents the interquantile range
from 10$^{th}$ and 90$^{th}$ percentile. Thus, the measurement site acts, on average, as a deposition area for the ultrafine and
accumulation particles, while it was a source area for greater size range (quasi-coarse mode). In order to explore the
behaviour of deposition events on the measurement site, only positive values of deposition velocity will be taken in



consideration, studying $V_d^+$. Median $V_d^+$ for UFP range was 0.90 mm s$^{-1}$ (IQR$_{10\text{-}90}$ 0.18 to 3.89 mm s$^{-1}$) , in the ACC mode median $V_d^+$ was 0.62 mm s$^{-1}$ (IQR$_{10\text{-}90}$ 0.16 to 2.38 mm/s) and in the CRS mode $V_d^+$ was 4.42 mm s$^{-1}$ (IQR$_{10\text{-}90}$ 1.57 to 13.09 mm s$^{-1}$). Also in this case $V_{ex}$ and $V_d^+$ analyses were separated in three periods (with/without snow and melting). In particular, $V_d^+$ increases from the snow surface to the snow-free conditions in all size range. It increased from a median value of 0.73 mm s$^{-1}$ (SC) to 1.14 mm s$^{-1}$ (SF) for UFP particles. In the same way, $V_d^+$ increased from

SC to SF phase with a maximum during SM period for ACC and CRS size range (Table 5).

**Table 4**. Statistical parameters for exchange velocity ($V_{ex}$), deposition velocity ($V_d^+$) and deposition velocity normalized for the friction velocity ($V_d^+/u_*$) calculated on the whole measurement period, separating the UFP, ACC and CRS particle size ranges.


| $V_{ex}$ (mm s$^{-1}$) | UFP | ACC | CRS |
|---|---|---|---|
| D$_{gm}$ (µm) | 0.035 | 0.42 | 1.45 |
| 10$^{th}$ prc. | -2.00 | -2.02 | -11.25 |
| mean | -0.02 | -0.24 | -1.33 |
| std.dev | 12.53 | 3.36 | 12.62 |
| std.err | 0.17 | 0.05 | 0.19 |
| median | 0.34 | 0.10 | -1.36 |
| 90$^{th}$ prc. | 2.82 | 1.53 | 8.48 |

| $V_d^+$ (mm s$^{-1}$) | UFP | ACC | CRS |
|---|---|---|---|
| D$_{gm}$ (µm) | 0.035 | 0.42 | 1.45 |
| 10$^{th}$ prc. | 0.18 | 0.16 | 1.57 |
| mean | 1.99 | 1.08 | 6.28 |
| std.dev | 5.76 | 1.55 | 6.62 |
| std.err | 0.10 | 0.03 | 0.15 |
| median | 0.90 | 0.62 | 4.42 |
| 90$^{th}$ prc. | 3.89 | 2.38 | 13.09 |

| $V_d^+/u_*$ (10$^{-3}$) | UFP | ACC | CRS |
|---|---|---|---|
| D$_{gm}$ (µm) | 0.035 | 0.42 | 1.45 |
| 10$^{th}$ prc. | 1.7 | 1.5 | 11.2 |
| mean | 14.5 | 8.8 | 54.7 |
| std.dev | 37.1 | 32.0 | 71.4 |
| std.err | 0.6 | 0.7 | 1.7 |
| median | 6.9 | 5.1 | 35.9 |
| 90$^{th}$ prc. | 28.7 | 18.2 | 108.9 |

The observed deposition or exchange velocity is in good agreement with previous measurements on snow and ice in the polar regions: however, a comparison with measured deposition velocities in the literature is a challenge because work on ice/snow surfaces is very seldom reported, and often different instruments and methods (such as physical

sampling, gradient method, eddy covariance) are used. Measurements reported by Duann et al. (1988) on snow gave





an average $V_d^+ = 0.34 \pm 0.14$ mm s$^{-1}$ for particles in the size range between 0.15 and 0.5 µm (Fig. 8a). In the high Arctic, Nilsson and Rannik (2001) report a mean deposition velocity $V_d^+ = 1.4$ mm s$^{-1}$ over ice in the nucleation mode and $V_d^+ = 0.51$ mm s$^{-1}$ in the Aitken mode (with a size distribution peaking at about 60-70 nm).

**Table 5**. Statistical parameters for the exchange velocity ($V_{ex}$), deposition velocity ($V_d^+$) and deposition velocity normalized for the friction velocity ($V_d^+/u_*$) calculated on snow (SC), snow melting (SM) and snow-free (SF) periods, separating the UFP, ACC and CRS particles size range.

| $V_{ex}$ (mm s$^{-1}$) | UFP | | | ACC | | | CRS | | |
|---|---|---|---|---|---|---|---|---|---|
| | SC | SM | SF | SC | SM | SF | SC | SM | SF |
| 10th prc | -2.40 | -1.85 | -1.54 | -1.42 | -2.69 | -2.92 | -9.30 | -13.71 | -13.12 |
| mean | -0.98 | 0.95 | 0.95 | -0.33 | -0.04 | -0.25 | -1.85 | -0.30 | -1.35 |
| std.dev | 16.8 | 6.13 | 4.24 | 3.47 | 3.63 | 2.60 | 12.56 | 13.60 | 11.38 |
| std.err | 0.33 | 0.18 | 0.11 | 0.07 | 0.10 | 0.08 | 0.27 | 0.41 | 0.38 |
| median | 0.22 | 0.52 | 0.48 | -0.05 | 0.27 | 0.12 | -1.54 | 1.23 | -1.15 |
| 90th prc | 1.85 | 3.97 | 3.81 | 0.88 | 2.52 | 2.17 | 5.87 | 13.01 | 10.42 |

| $V_d^+$ (mm s$^{-1}$) | UFP | | | ACC | | | CRS | | |
|---|---|---|---|---|---|---|---|---|---|
| | SC | SM | SF | SC | SM | SF | SC | SM | SF |
| 10th prc | 0.15 | 0.20 | 0.20 | 0.14 | 0.23 | 0.23 | 1.34 | 2.31 | 1.41 |
| mean | 1.51 | 2.56 | 2.29 | 0.66 | 1.63 | 1.41 | 4.27 | 9.16 | 7.14 |
| std.dev | 6.41 | 6.07 | 4.17 | 1.16 | 1.97 | 1.48 | 3.14 | 9.36 | 6.61 |
| std.err | 0.16 | 0.21 | 0.13 | 0.03 | 0.07 | 0.07 | 0.10 | 0.39 | 0.32 |
| median | 0.73 | 1.21 | 1.14 | 0.42 | 1.02 | 0.94 | 3.50 | 6.48 | 4.82 |
| 90th prc | 2.62 | 5.41 | 5.06 | 1.29 | 3.62 | 3.04 | 7.91 | 17.50 | 16.81 |

| $V_d^+/u_*$ ($10^{-3}$) | UFP | | | ACC | | | CRS | | |
|---|---|---|---|---|---|---|---|---|---|
| | SC | SM | SF | SC | SM | SF | SC | SM | SF |
| 10th prc | 1.7 | 1.7 | 1.8 | 1.4 | 1.7 | 1.8 | 11.3 | 11.1 | 10.9 |
| mean | 12.6 | 15.8 | 16.3 | 7.3 | 10.0 | 10.9 | 44.9 | 70.8 | 56.0 |
| std.dev | 38.4 | 43.7 | 28.0 | 42.5 | 11.6 | 16.1 | 43.2 | 102.5 | 69.5 |
| std.err | 1.0 | 1.6 | 0.9 | 1.2 | 0.5 | 0.8 | 1.5 | 4.5 | 3.6 |
| median | 6.4 | 7.1 | 7.4 | 4.0 | 6.7 | 7.0 | 33.0 | 45.2 | 34.7 |
| 90th prc | 24.0 | 29.1 | 36.7 | 13.3 | 21.7 | 23.3 | 91.0 | 141.5 | 116.2 |

Gronlund et al. (2002) reported a median $V_d^+$ of about 3.3 mm s$^{-1}$ (and an interval between 0.8 and 18.9 mm s$^{-1}$) over snow for total particles larger than 10 nm. The values reported by this last work are significantly larger than that observed in the present data set. Held et al. (2011.a) observed median $V_d^+$ values ranging from 0.27 to 0.68 mm s$^{-1}$ during deposition-dominated periods. In Held et al (2011.b) a deposition velocity on the snow surface in the Arctic pack ice ranged from 0.28 to 0.58 ($\pm$ 0.4) mm s$^{-1}$ by eddy covariance. A fair agreement with observations from Contini

et al. (2010) at the Nanseen Ice Sheet (Antarctica) over ice/snow, where median $V_{ex} = 0.19$ mm s$^{-1}$ was observed for total particle larger than 10 nm. Further, Contini et al. (2010) observed a median deposition velocity $V_d^+ = 0.65$ mm s$^{-}$





[1]. In the Hudson Bay (Canada), Whitehead et al. (2012) measured a mean $V_{ex}$ of $0.12 \pm 0.11$ mm s⁻¹ on sea ice, for particle greater than 2.5 nm. Seasonal differences in deposition have also been observed in the cryosphere with greater values during warmer months (Macdonald et al., 2017), collecting snow samples on average every 4 days at Alert, Nunavut (Arctic Canada). In Ibrahim et al (1983) a deposition velocity between 0.39 and 0.96 mm s⁻¹ over snow surface was observed for 0.7 µm diameter particles using artificial collectors for liquid scintillation counting technique.

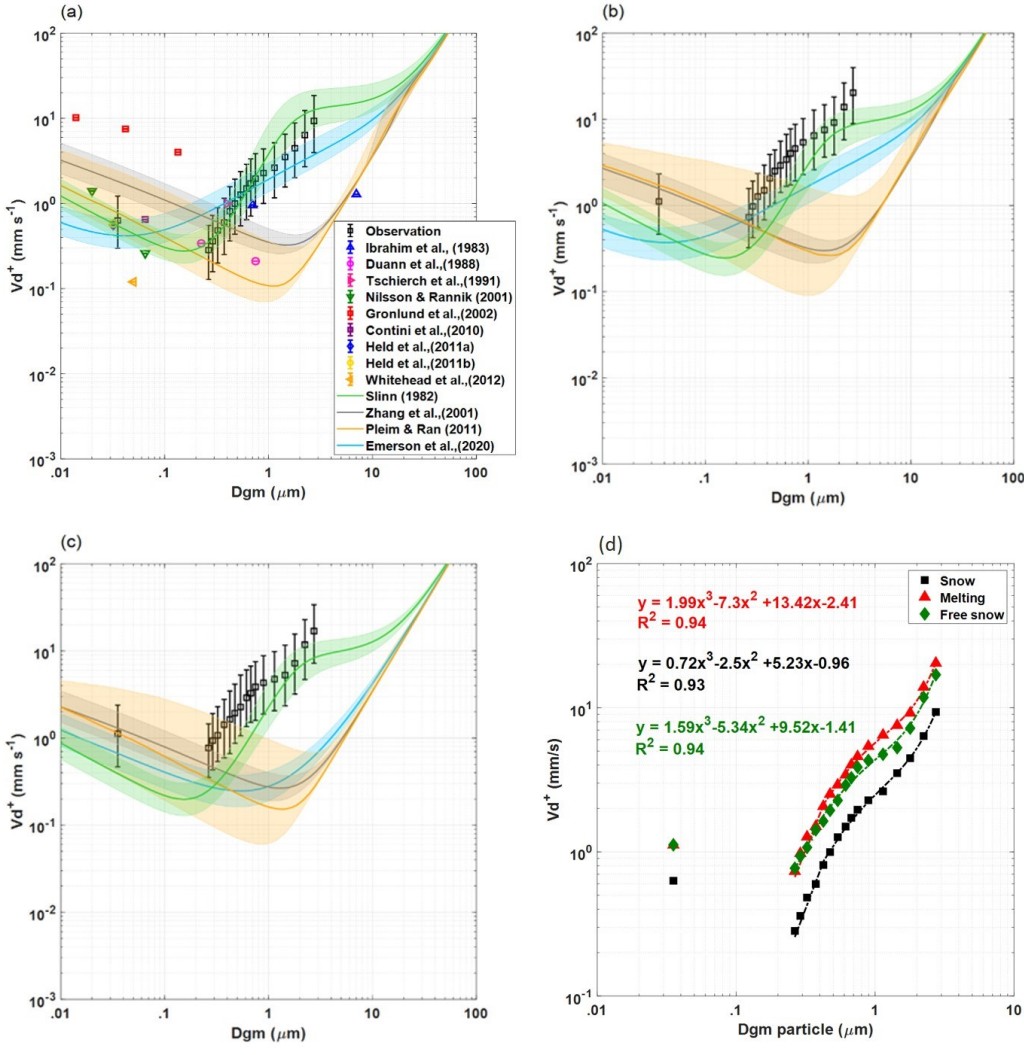

**Figure 8.** Median $V_d^+$ as a function of the geometric mean diameter measured during a) snow cover, b) melting period, and c) snow-free phase. A comparison with the model predictions is also reported in each panel. Observations are shown in symbols, models in lines. Error bars and shaded areas represent the interquartile range. In a) panel, median measured $V_d^+$ are compared to previous measurements of deposition velocities over snow. d) Functional fit for $V_d^+$ depending on the geometric mean particle diameter for the different period analysed in this work.



Even if the particle sizes range were not exactly the same, the $V_{ex}$ (1 mm s$^{-1}$) reported by Tschiersch et al. (1991) are still comparable with values measured at the closest size range (Fig. 8a). For a more detailed and comprehensive review of deposition velocities see Pryor et al. (2008), Whitehead et al. (2012), Saylor et al. (2019); Emerson et al. (2020), Farmer et al. (2021). From the graphs in Fig. 8, $V_d^+$ (median values) increases with particle diameter, and $V_d^+$ is greater in all size bins during the melting phase with respect to the other periods, due essentially to the increased

ability of the water surface to capture the particles (in all size ranges) by interception and impaction mechanisms. $V_d^+$ for the snow surface coverage results the lowest. To put our observations into context, deposition velocities observations from previous studies have been plotted in Fig. 8, alongside four models for dry deposition velocity developed by Slinn (1982, S82), Zhang et al., (2001, Z01), Pleim and Ran (2011, PR11) and Emerson et al. (2020, E20). Dry deposition models were based on a custom code (DepoBoxTool) translated in MATLAB (ver. 2018b) from

the original form in Python (available at: https://doi.org/10.5281/zenodo.4749548, accessed on September 26, 2022) (Shu et al., 2021a; Shu et al., 2021b). Figure 8a shows the deposition velocity measured and predicted during the snow period, whereas Figs. 8b and 8c display the results during melting and snow-free period, respectively. Further, it seems that $V_d^+$ observed in this work follow the shape of the Slinn model output. As reported by Saylor et al. (2019) for smooth surfaces such as water or snow/ice, models provide a minimum deposition rate for particles in accumulation

mode. Data suggest a minimum in the 0.08 – 0.15 μm range, with a strong increase in $V_d^+$ above 0.25 μm, rather than the gradual decrease predicted by some of the algorithms (i.e Z01, PR11). Our results are in line with the observations of E20 revisiting the importance of interception processes with respect to Brownian diffusion. Apparently, interception can be a significant term of particle loss even over ice/snow surfaces, whose actual structures and fine irregularities contrast with the idealized "smoothness" traditionally assumed for this land type.

The selected deposition velocity models take in input our meteorological and micrometeorological observations (temperature, pressure, relative humidity, L, $u_*$), situ-specific surface properties (roughness length, canopy height, displacement height, land use category) and particle properties (density and diameter). Note that most model outputs are highly sensitive to canopy structure parameters and friction velocity. It is worth considering, for example, the high sensitivity of PR11 to the value of the convective velocity scale, which can vary significantly with the surface

roughness defined for each land use type (Saylor et al., 2019). On the other hand, Z01 produces the widest differences in predicted deposition velocities among land use types, showing weak deposition velocities compared to other models especially in the accumulation and coarse mode. The comparison of predictive models with our observations shows a good agreement, especially with S82, which seems to better fit the data over a broader range of particle diameters, properly predicting a minimum in a diameter range of 0.1 - 0.3 μm, as well as the deposition velocities for

accumulation and coarse mode. On the contrary, the Z01 and PR11 models, show the minimum around 1 μm, providing a wide underestimate of the deposition velocity for particles above 0.3 μm. Specifically, during the snow period (Fig. 8a), the considered models well predict deposition rates for particles below 0.1 μm, with the exception of Z01, which overestimates $V_d^+$ in that range. By contrast, a slight and wide underestimation of $V_d^+$ above 0.3 μm was observed for E20, Z01, and PR11 parametrizations. During the melting period (Fig. 8b), a good agreement was

observed in the 0.03–0.3 μm size range, although slight under prediction was observed for S82 and E20. For particle diameters above 0.3 μm, a wide underestimation occurs for all parameterizations except S82, which shows a good




agreement especially in the particle range above 1 μm. Finally, during the snow-free period, an overestimation of the deposition velocity for ultrafine particles was observed in Z01 and PR11, which remained in agreement with measurements up to 0.3 μm particles. AlthoughS82 and E20 models show an underestimation of $V_d^+$ for ultrafine

particles, a good agreement was observed for accumulation and coarse mode, especially for S82. In view of this results, most analysed parameterizations show an underestimation of the deposition velocity greater than an order of magnitude in the range of 0.2-0.3 μm, except for S82. In general, due to the sensitivity of the models to input parameters, the differences between model outputs and our data should be considered with caution. However, our data clearly show that minimum deposition velocity is reached around 0.1 μm of particle diameter, that $V_d^+$ values

monotonously increase over the particle size interval 0.1 to 10 μm, and our data support the parameterizations in the high range of $V_d^+$ for this particle size interval.

**3.5 Particle deposition velocity parameterization**

Regarding the trend of the deposition velocity as a function of particles diameter, $V_d^+$ can be described by a third order polynomial function of mean geometric diameter in a typical saddle-shaped graph with an increase in magnitude

toward the lowest and highest side of the size spectrum and a minimum in the Aitken mode. A robust (bi-square) polynomial fit to our observations was computed for the three considered periods, with a resulting goodness of fit $R^2$ and a root mean square error (RMSE) as reported in Table 6.

$$V_d^+(D_{gm}) = p_1(D_{gm})^3 + p_2(D_{gm})^2 + p_3(D_{gm}) + p_4 \tag{3}$$

where p-values in (3) denote the polynomial fitting coefficients.


**Table 6**. Polynomial fit parameters and goodness of fit for each observed curve.

| $V_d^+$ | $p_1$ | $p_2$ | $p_3$ | $p_4$ | $R^2$ | RMSE |
|---|---|---|---|---|---|---|
| SC | 0.72 | -2.5 | 5.23 | -0.96 | 0.93 | 4.02 |
| SM | 1.99 | -7.3 | 13.42 | -2.41 | 0.94 | 3.29 |
| SF | 1.59 | -5.34 | 9.52 | -1.41 | 0.94 | 1.93 |
| $V_d^+/u^*$ | | | | | | |
| SC | $6\cdot10^{-5}$ | $-2\cdot10^{-4}$ | $5\cdot10^{-4}$ | $-8\cdot10^{-5}$ | 0.93 | $2.2\cdot10^{-4}$ |
| SM | $3\cdot10^{-4}$ | $-1\cdot10^{-3}$ | $2\cdot10^{-3}$ | $-3\cdot10^{-4}$ | 0.94 | $6.7\cdot10^{-4}$ |
| SF | $3\cdot10^{-4}$ | $-1\cdot10^{-3}$ | $2\cdot10^{-3}$ | $-3\cdot10^{-4}$ | 0.94 | $4.5\cdot10^{-4}$ |

Deposition velocity increased with friction velocity in all size ranges and several studies (Nilsson and Rannik 2001,

Pryor et al., 2008, Contini et al., 2010) found a positive correlation between $V_d^+$ and the friction velocity, in particular for $u_*$ > 0.1 m/s the relation can be described linearly. Our data confirm a linear correlation (see Fig. 9a), but the correlation coefficient varies considerably with particle size, with an angular coefficient $m = 1.27$ ($R^2$=0.9) for UFP, $m = 0.78$ ($R^2$=0.67) for ACC and finally, $m = 7.15$ ($R^2$=0.86) for CRS. In this context, a reasonable normalization of the deposition velocity is the ratio $V_n = V_d^+/u_*$ to obtain a better comparability between different studies (Table 5).





An analogous fit for normalized $V_n$ resulted in a similar goodness of fit and respective fitting coefficients reported in Table 6, segregated for the presence or not of the snowpack.

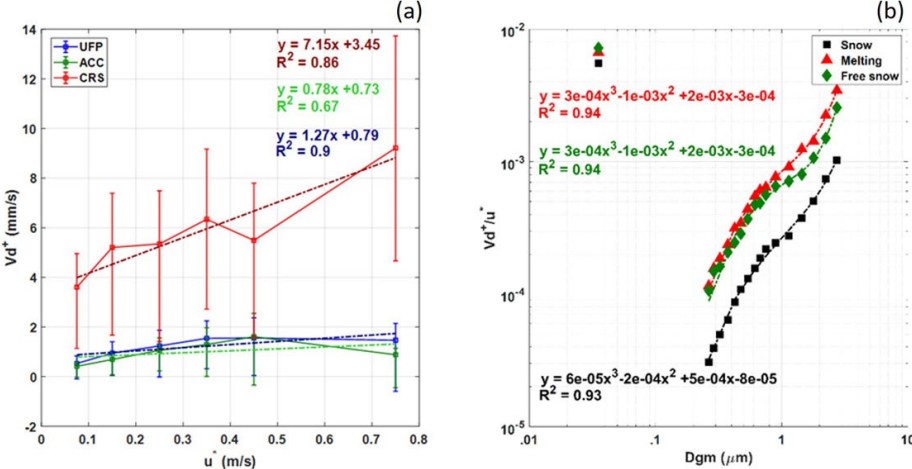

**Figure 9.** a) Relation between median $V_d^+$ and friction velocity for the different size ranges. Vertical bars represent the interquartile range of the deposition velocity within the specific interval of u∗. Friction velocity intervals were

selected to optimize the number of data points within each interval and, hence, provide a statistically reliable median deposition velocity. b) Functional fit for $V_n$ as a function of the geometric mean particles diameter for the different period analysed in this work.

## 4 Conclusions

Aerosol fluxes were measured in the Arctic site of Ny-Ålesund (West Spitzbergen, Svalbard). The measurement

campaign was carried out between March and August 2021. Number concentrations for ultrafine particles increased from March to August, with new-particle formation processes probably contributing to summertime concentrations, boosted by the enhanced biological emissions in warm months. By contrast, number concentration for larger-sized particles shows the maximum concentration in March, during the Arctic haze season, and then decreasing to a minimum concentration in June. Turbulent fluxes in the ultrafine particle size range resulted prevalently downwards

especially in summertime when concentrations are highest. Particles fluxes in the accumulation and quasi-coarse mode were positive (upward) during the colder months, indicating that the site behaves on average as a source for particles, even if these fluxes were very small. In the warmer months, particles fluxes were slightly negative, showing that the area in exam acts as a weak sink for particulate in those size ranges. The overall median deposition velocity ($V_d^+$) values were 0.90, 0.62 and 4.42 mm s⁻¹, for UFP, ACC and CRS, respectively. A more pronounced deposition was

observed, on average, during melting and snow-free periods over the whole dimensional range, while the median deposition velocity ($V_d^+$) with the snowpack was of 0.73, 0.42 and 3.50 mm s⁻¹. Our results indicate a low sensitivity of the saddle-shape of deposition velocity (as a function of the geometric mean diameter) to the surface characteristics, showing a fixed minimum in the range of 0.1 - 0.2 μm and an increase in deposition velocity for particles > 0.3μm. The increase in deposition velocity is probably due to the increase in roughness length, due to the presence of obstacles,





such as rocks or irregularities in the snow cover (and on the bare soil after melting), that perturb the flow with an increase in deposition. The observed deposition velocity depends on friction velocity, therefore, it is valuable to normalize deposition velocity with the $u_*$ in order to obtain a better comparison between different measurement sites. The median normalized velocity on the whole period ($V_n$) was $6.9 \cdot 10^{-3}$, $5.1 \cdot 10^{-3}$, and $35.9 \cdot 10^{-3}$ for UFP, ACC and CRS, respectively. In general, our observations of deposition velocities are aligned to previous literature in similar

environments (i.e. ice/snow), especially for particles in the size range 0.01-1μm, with a percentage differences lower than 50% of magnitude. The behavior of $V_d$ as a function of particle diameter resulted well predicted by the Slinn (1982) parameterization. The Slinn-derived models considered in this work provide different predictions with similar land covers (snow, tundra and water ponds) and same environmental conditions, as reported also by Saylor et al., (2019). However, large discrepancies between modelled and observed deposition velocities for the accumulation mode

were observed for most of the models used (i.e. Z01, PR11 and E20). The predictions of the available parameterizations generally do not agree with the observations, especially on snow/ice surfaces, and the discrepancy in deposition velocity values can be as high as two orders of magnitude. The polynomial parameterization proposed in this study fits our observations and the analyzed models which properly represents the size-dependence and magnitude of deposition velocity. The fit function can be implemented in existing chemical transport models, taking

into account the role of turbulence on dry deposition, which is a process typically neglected in regional and global models. Overall, our findings indicate that the most common parameterizations for dry depositions in polar areas could contribute significantly to the overall uncertainty in global models simulating SLCFs climate effects (Whaley et al., 2022). The scarce measurements on the cryosphere, and the consequent large gap in deposition velocity modeling demonstrate the need to improve the understanding of deposition processes in polar areas.

*Data Availability:* Data is available upon request

     *Author contribution:* AD and GP planned the experimental design and guided the research. AD, GP, MM, FS collaborated to data collection and post-processing. AD, GP and DF carried out the analysis presented in this paper and drafted the manuscript with contributions from all co-authors. AD, GP and DF developed the code used to analyze the data. AD and SD managed and provided the funding projects. All authors reviewed and edited the manuscript,

cooperating to interpretation of results, wrote, read, commented, and approved the final manuscript.

     *Competing interests:* The authors declare no competing interests.

     *Financial Support:* This work has been funded and conducted in the framework of the Joint Research Center ENI-CNR - "Aldo Pontremoli" within the ENI-CNR Joint Research Agreement, WP1 "Impatto delle emissioni in atmosfera sulla criosfera e sul cambiamento climatico nell'Artico". Further, this work was partially financed by the Svalbard

Science Forum for the Arctic Field Grant "Aerosol Flux in Arctic" (ALFA) project (RiS ID 11390; NFR contract 310658).

     *Acknowledgements*



The authors acknowledge Fabio Massimo Grasso (ISAC-CNR) for his help in setting up the instrumental acquisition system before the measurement campaign. Authors acknowledge the Institute of Polar Science (ISP-CNR) and its staff for the logistics of the Arctic Station "Dirigibile Italia" in Ny-Ålesund. We would like also to acknowledge dr. Nuncio Murukesh of the Indian Arctic Program for the precipitation data and Rita Traversi of University of Florence for the SMPS data used in this work.

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
