# Peer review of "Characterization of size-segregated particles turbulent flux and deposition velocity by eddy correlation method at an Arctic site"

_Atmospheric Chemistry and Physics, 2022_

## Author Comment (AC1)

**Characterization of size-segregated particles turbulent flux and deposition velocity by eddy correlation method at an Arctic site. A. Donateo et al. - Responses to #RC1.**

We thank the Reviewer for her/his comments and feedback and for taking the time to help improve this manuscript. Reviewer comments are presented in blue, and sections that have been added to the text are coloured green. Author's responses are reported in black. Note: Figures in the manuscript are referred to as "Fig. X", figures included in these review responses, but not in the manuscript are referred to as "R.Fig. Y".

**General Comments**

This paper presents 5 months of particles fluxes measured by eddy covariance at Ny-Alesund, Svalbard. Deposition velocities as a function of particle size are derived for snow covered, melting and snow-free conditions. In addition to the ultra-fine particle numbers obtained by a condensation particle counter, the 16 size channels of a optical particle counter are summed into 2 broader size classes in order to obtain workable statistics. An analysis is performed first on all fluxes regardless of direction, and downward fluxes are then analysed separately to construct deposition velocity parameterizations for the three periods. Results are compared to previously published numbers and models and agree reasonably well.

There are some issues with this analysis that cannot be improved upon after the fact, such as the measurement site on top of a building that is likely to cause flow conditions less than ideal for eddy covariance, or the limited spectral response of the particle instrumentation. These should be discussed and justified in more detail than is currently the case. There are also a few issues that may warrant some additional thought and perhaps some further analysis. Simply concentrating on only positive (deposition) fluxes without accounting for natural scatter across the zero line is somewhat arbitrary and is likely to introduce artificial biases towards larger numbers; therefore, this should be justified in more detail. Some additional information, such as an ogive of particle fluxes and some statistics on the upward (emission) particle fluxes, either in the manuscript or a supplement, would also be helpful to convince the reader of the validity of the results.

Given the paucity of published direct measurements of particle fluxes, and especially in the Arctic, this paper can represent a useful addition to the field if these issues are addressed.

**Specific Comments**

Line 104: Please include a photo of the building and mast, and state the dimensions of the building (height, footprint) and the mast. Doing flux measurements on a rooftop can cause all kinds of problems due to flow distortion, and for some directions more than others, but a bit more on that later.

The mast is on the rooftop of Gruvebadet building. The mast is 4 m tall above roof level (R.Fig.1a). The building is 5 m high (above ground level) (R.Fig.1b). The anemometer sensing volume and head of the inlet is height 60 cm and 50 cm, respectively. The total height above the ground level is 9.60 m (in the manuscript is reported as "about 10 m", now "9.6 m"). The short side of the building is 12 m, and the long side is 22 m (R.Fig.1c). The mast is located very close to roof edge (northeast-facing side). Possible flow distortions could arise for the presence of the rooftop in the wind direction sector between 180° (S) and 240° (SW) (see R.Fig.1c – yellow sector). Winds coming from this sector is uncommon in this measurement campaign (about 5% of quality assured data) (Fig.3f). Furthermore, the momentum flux for the roof wind sector (see above) did not show positive anomalies (R.Fig.2). An analysis has been performed on spectra for u, v, and w wind components from the roof sector (R.Fig.2b). From this graph can be noted that both the vertical and the horizontal wind components follow very well the trend line (black dashed line) at -5/3 in the inertial subrange, without any evident anomaly. Thus, it doesn't seem to be any important flow distortions arising from the roof sector. For these reasons the Authors decided to take in consideration all the quality assured dataset, also including the wind sector corresponding to the roof area. In the new version of the paper, Authors decided to produce a Supplementary material file. Thus, R.Fig.1 was inserted in this Supplementary file as Fig.S1.

[Figure]

[Figure]

[Figure]

R.Fig.1. (a) Picture of the EC mast located on the roof of Gruvebadet lab and of (b) a front view of the building. (c) A sketch of the top and front view of Gruvebadet lab and EC setup.

[Figure]

R.Fig.2. (a) Momentum flux data points measured in the wind sector between 180° and 240° N. Please, note that the data points are not consecutive in time. (b) Normalized median spectra for u, v and w wind components as a function of the normalized frequency $f_n$ selected from the roof sector. Black dashed line represents the slope theoretically predicted in the inertial subrange.

L150: If T comes from the sonic (Ts), it's not the actual temperature, but rather a virtual temperature (or close to virtual, for all intents and purposes) which also depends on moisture. Therefore, what you are calculating is not strictly the sensible heat flux H, but a virtual sensible heat flux that is probably close enough to H for all practical purposes. But it's best to be exact at the definition phase.

The Authors agree with the Reviewer. In the revised version of the manuscript the exact definition is now reported as virtual heat sensible flux.

L155 and after: make the "i" a subscript

The formula has been corrected.

L157: here you define Vdi, but later you only use Vex. Make this consistent throughout. Vex is more transparent and therefore a better choice.

The formula has been corrected.

Eq. (2): should include Ts rather than T

The equation has been corrected according to the Reviewer's suggestion.

Section 2.4 could use some rereading and grammatical clean-up

This Section has been rewritten and deeply corrected in the revised version of the manuscript as follow:

Half-hour periods were hard flagged for dropouts, discontinuities caused by power loss, or values outside the absolute limits, and discarded from the dataset. These events resulted in 25% and 24% of data being rejected for the CPC and OPC fluxes, respectively. The OPC measurements were discontinued on July 29th due to technical issues. The raw data were pre-processed by first applying a despiking procedure to eliminate spikes caused, for example, by electronics issues. Spikes in the 100 Hz (anemometer) and 1 Hz (CPC and OPC) time series were removed from the dataset and replaced by linear interpolation using a procedure described by Vickers and Mahrt (1997). A spike detection algorithm was applied to the raw high-frequency data defining spikes as absolute deviations from the mean of a threshold value, that is 6-fold $\sigma$ (where $\sigma$ is the variance of 10 min sub-interval). Using a closed path instrument (i.e., CPC or OPC) with a sampling tube, air sampled by the sonic anemometer takes many seconds to reach the scalar sensor, leading to a time lag between the vertical wind component fluctuations $w'$ and concentration fluctuations $c'$. Without correcting for the time delay, vertical wind component fluctuations do not correlate properly with concentration fluctuations, leading to incorrect flux estimation. Such time lag was estimated by means of cross-correlation analysis by moving time series forward, defining the maximum covariance between the vertical component of wind speed and particle number concentration, considering a time window between 3 s and 5 s. This time window was estimated using the flow rate, tube length and response time (Deventer et al., 2015). A mean time lag of 4.14 s was calculated for CPC and 3.77 s for OPC (for all size channels) measurements, respectively. The wind data were rotated along the streamline reference system (McMillen, 1988) via three rotations. The first two rotations set a reference system that, for each averaging period, aligns the streamline velocity component with the mean wind velocity vector. The third rotation was performed except when the absolute value of the angle of attack (McMillen, 1988) was greater than 15° (about 3% of total cases). In neutral or stable atmospheric conditions with low wind speed, weak and intermittent turbulence (Sun et al. 2012, Schiavon et al., 2019), the sub-meso motions do not follow surface-layer similarity, but they can still contribute to observed statistics (Vickers and Mahrt 2006;

Liang et al., 2014; Mortarini et al., 2016). According to some authors (e.g., Vickers and Mahrt 2003), similarity relationships should be evaluated only after filtering out the contribution from these motions. In this work, the energy contributions related to non-turbulent, sub-meso motions, with time scales often longer than the investigated time window, were removed by a recursive digital filter both for heat and particles fluxes (Falocchi et al., 2018; Pappaccogli et al., 2022). The recursive digital filter worked on a different time scale, dependent on atmospheric stability conditions. Ogive analysis (Fig. 2b) was carried out in order to estimate a properly time scale (Metzger and Holmes, 2008). A detailed description of the methodology used for spectral and ogive analysis is reported in Pappaccogli et al. (2022). Time scale for unstable atmosphere is 522 s, whereas the value decreases to 350 s and 340 s for neutral and stable conditions, respectively. This filter does not introduce any phase shift or signal amplitude attenuation in the filtered time series. The filtering procedure used two data buffer (1800 s long), before and after the considered 30 min period of investigation. A fundamental assumption of the EC method is that fluctuations are statistically stationary during the chosen averaging time to ensure the calculation of an ensemble average. A stationarity tests as reported by Mahrt (1998) were carried out on sonic temperature and particles concentration fluctuations (Cava et al., 2014; Věcenaj and De Wekker, 2015). A lower detection limit for the fluxes in the sampling system was computed using the method proposed by Langford et al. (2015) and defined at 2.8 cm$^{-2}$ s$^{-1}$ for the CPC and 0.3 cm$^{-2}$ s$^{-1}$ for the OPC. To ensure the study of only particle dry deposition, all data corresponding to a precipitation intensity greater than 0.1 mm h$^{-1}$ for a time period greater than 5 min (on the averaging period of 30 min) were also rejected. Error associated to the random and limited statistical counting (relative error, %) was estimated through the approach reported in Deventer et al. (2015) for particle number concentration $\delta(N)$ and fluxes $\delta(w'N')$.

L192: rephrase/clarify. There was no correction for large angles, but the small ones were corrected?

This sentence referred to the data rotation procedure in the streamline reference system as recommended by McMillen (1988). In this work, the third rotation was not applied if the angle of attack was greater than 15°. In these cases (3% of total), only a double rotation was performed.

McMillen, R. T.: An eddy correlation technique with extended applicability to no simple terrain, Bound-Lay. Meteorol., 43, 231–245, 1988.

The sentence has been rephrased in the revised version of the manuscript as follow:

The third rotation was performed except when the absolute value of the angle of attack (McMillen, 1988) was greater than 15° (about 3% of total cases).

L194: please clarify.

The sentence has been rephrased in the revised version of the manuscript as follow:

In neutral or stable atmospheric conditions with low wind speed, weak and intermittent turbulence (Sun et al. 2012, Schiavon et al., 2019), the sub-meso motions do not follow surface-layer similarity, but they can still contribute to observed statistics (Vickers and Mahrt 2006; Liang et al., 2014; Mortarini et al., 2016).

L202: dependent on (not according to)

The sentence has been corrected in the manuscript.

Fig. 2:

- those slopes in the inertial subrange are not even close to -7/3, except for the highest 3 points perhaps! I would temper that assertion in L257.

Authors agree with the Reviewer. In the original figure (Fig.2a) the kinematic heat flux has been calculated on the decimated time series (w and $T_s$ at 1 Hz) to obtain the same frequency of CPC and OPC (that is, 1 Hz). However, in this way the heat flux did not clearly show its real trend. So, we decided to change Fig.2a, reporting the three cospectra, each with its normalized frequency. At the same time, the data binning has been modified in order to better represent the cospectral results. Now, in this new figure, it is clearer the -7/3 trend of the scalar cospectra. CPC and OPC cospectra show their losses at high frequencies, as expected, due to their low time response.

[Figure]

**Figure 2**. (a) Normalized median cospectra of kinematic heat flux (blue), UFP (red) and ACC-CRS particle flux (green) as a function of $f_n$. The binned median cospectra (about 1500) were computed from continuous period of 1h. Black dashed line represents the slope theoretically predicted in the inertial subrange (Kolmogorov, 1941); (b) median normalized ogives for kinematic heat flux as a function of natural frequency (Hz) for the three classes of atmospheric stability.

- In a), the particle flux signal drops by 5 orders of magnitude over the frequency range shown whereas the heat flux signal drops by only 3.5 orders of magnitude. Please discuss what this means

in terms of underestimating fluxes associated with smaller eddies, and how that may affect the integrated total flux.

As said previously, the CPC and OPC cospectra have high frequencies losses due to their low sampling frequency (1 Hz) and their low first order frequency response. As clearly stated in the manuscript (L278-L286) the high frequency losses for CPC and OPC have been corrected using a parametric/in situ procedure proposed by Horst (1997). According to this frequency correction an average loss of about 30% and 21% has been calculated for CPC and OPC, respectively (L282-L283).

Horst, T.W.: A simple formula for attenuation of eddy fluxes measured with first order-response scalar sensor. Boundary-Layer Meteorol. 82, 219–233, https://doi.org/10.1023/A:100022913, 1997.

- Showing heat fluxes in b) is fine, but it would be more insightful to show the ogives for the particle fluxes as well. Obviously, these will be significantly noisier, but showing that these look reasonable will go a long way towards convincing the reader that they behave properly.

In R.Fig.3 the ogives for the CPC and OPC data, for the whole dataset (with no distinction for stability conditions). As stated from the Reviewer, these ogives are significantly noisier as also reported in other works (Held et al, 2011; Pappaccogli et al, 2022). In our work, the ogives have been used to calculate the time scale for the application of the digital filter starting from the kinematic heat flux. We believe that adding CPC and OPC ogives to Fig.2b could potentially confuse the Reader. However, R.Fig.3 was inserted in the Supplementary material file as Fig.S4.

[Figure]

R.Fig.3. Median normalized ogives for UFP flux (red) and ACC-CRS particle flux (green).

Held, A., Brooks, I. M., Leck, C., and Tjernström, M.: On the potential contribution of open lead particle emissions to the central Arctic aerosol concentration, Atmos. Chem. Phys., 11, 3093 - 3105, https://doi.org/10.5194/acp-11-3093-2011, 2011.

Pappaccogli, G., Famulari, D., and Donateo, A.: Impact of filtering methods on ultrafine particles turbulent fluxes by eddy covariance, Atmos. Environ., 285, 119237. https://doi.org/10.1016/j.atmosenv.2022.119237, 2022.

- You could easily clean up the neutral ogive by widening the z/L range definition for "neutral".

We tried what suggested by the Reviewer, and found that widening the z/L range for neutral atmospheric conditions from the present range (from -0.01 to 0.01) to a larger range (i.e. from -0.05 to 0.05) the number of ogives considered did not change considerably, and further subtracted cases from the unstable and stable condition (that are the most representative for the dataset). We believe the best option is to stick to the originally defined range, between -0.01 and 0.01 chosen to identify the neutral atmosphere, as also found in other works (Nordbo et al, 2013; Lindroth et al, 2022).

Lindroth, A., Pirk, N., Jónsdóttir, I.S., Stiegler, C., Klemedtsson, L., Nilsson, M.B., 2022. CO2 and CH4 exchanges between moist moss tundra and atmosphere on Kapp Linné, Svalbard. Biogeosciences 19, 3921–3934. https://doi.org/10.5194/bg-19-3921-2022

Nordbo, A., Järvi, L., Haapanala, S., Moilanen, J., and Vesala, T.: Intra-city variation in urban morphology and turbulence structure in Helsinki, Finland. Boundary-Layer Meteorol. 146(3), 469–496. https://doi.org/10.1007/s10546-012-9773-y, 2013.

L288: "below 0C" would be more accurate than "about -10C".

We thank the reviewer for this remark, in the revised version of the manuscript, we have replaced the old sentence giving an exact value of air temperature in the months of March and April, as follow:

During March and April, the air temperature was on average -7 °C.

Fig. 3: wind directions look better when plotted using dots rather than lines

The Authors agree with the Reviewer. Fig.3 has been revised replacing the previous panel d) with the time series of wind direction, with a wind rose (now located on panel f) representing better the wind direction characteristics.

[Figure]

**Figure 3.** Time series of the principal meteorological variables measured during the campaign in Ny-Ålesund. (a) air temperature (°C), (b) relative humidity (%), (c) wind velocity (m/s), (d) snow depth (m), (e) daily cumulative precipitation (mm), and (f) wind direction (degree N). In (d) panel three colour bands were used to separate the period characterized by snow cover (dark cyan) from melting snow period (cyan) and, finally, snow-free phase (light cyan).

Fig. 5: averaging together the whole measurement period, consisting of vastly varying climatic and heat flux regimes, may not be the best way to do this. I would suggest splitting into at least 2 panels, one for snow-covered and one for snow-free, plus perhaps one for the transition period.

The Reviewer has a good point here, and we modified Fig.5: it has now been revised to show the daily distribution of stability in the three different periods.

[Figure]

**Figure 5.** Percentage of atmospheric stability over the daily cycle for (a) snow cover, (b) melting and (c) snow-free period. Stability classes based on threshold values of the stability parameter ζ were defined as: very unstable (dark red, ζ< - 1) and unstable (red, ζ < - 0.01), neutral (green, - 0.01 ≤ ζ ≤ 0.01), stable (blue, ζ > 0.01) and very stable (dark blue, ζ > 1).

Also, a discussion about this aspect has been reported in the revised manuscript as follow:

As it can be noted in Fig. 5, stable conditions prevailed during the snow cover period (Fig.5a), with 62% of cases (and 26% very stable), while unstable stratification was more frequent in summer months (Fig.5c) (45% and 43% for unstable and very unstable, respectively). Neutral stratification occurred with a maximum frequency of 7% during the melting period, with no diurnal variability.

L337: according to the flux footprint estimate in Fig. 1, and the stated maximum footprint in L356, the ocean is well outside the footprint, and therefore it seems very unlikely that swell would be the cause of these upwards momentum fluxes. Could it be katabatic winds coming down Zeppelin Mountain? Or a very shallow sea breeze? Would need to look at wind directions carefully.

We thank the Reviewer for this remark: after this comment, we plotted momentum flux for a check, and displayed below here. Looking at the graph of the momentum flux time series (R.Fig.4) as a function of the wind direction (on colormap), a particular event can be observed (first red circle from left), with positive values of momentum flux. This positive peak corresponds to a very well-defined wind sector W (from 250° to 270°), that is, air masses coming from the glacier Brøggerbreen, SW of our measurement site, with relatively high velocity (on average 8 m s$^{-1}$). The same can be said for the second (in time) red circle event in R.Fig.4. Other events (blue circles) correspond to air masses coming from E-SE, that is from the Kongsfjiorden, most likely due to the effect of breeze swell, as stated, also by the Reviewer. R.Fig.4 was inserted in the Supplementary material file as Fig.S5.

[Figure]

R.Fig.4. Momentum flux for the measurement campaign reported as a function of wind direction (°N). Red circles highlight large positive momentum flux events, while blue ones indicate weaker events raising from E-SE air masses.

A discussion about this aspect has been added, and reported in the revised manuscript as follow:

The kinematic momentum flux was typically downward (Fig. 4b) with an average of -0.034 m$^2$ s$^{-2}$. However, for about 16% of the data, the direction of the momentum flux showed positive values. A particular event can be observed (Fig.S4) on 2$^{nd}$ June from 14:00 to 18:30, with large positive values of momentum flux (on average 0.40 m$^2$ s$^{-2}$). This event was characterized by air masses coming from the glacier Brøggerbreen, located W (250° - 270°) of our measurement site,

with relatively high velocity (on average 8 m s$^{-1}$). More frequent positive momentum flux events with lower intensity occurred, coinciding with air masses coming from E-SE (from the Kongsfjiorden), most likely due to the effect of a very shallow sea breeze swell (Grachev and Fairall, 2001).

L349: these are very large roughness lengths for snow or tundra surfaces, an order of magnitude higher than expected, suggesting that the building is introducing a significant roughness element to the flow observed at the sonic.

The Reviewer is right: checking our calculations for $z_0$, we realized that the measures were reported in centimetres and not meters. Now, all the values are reported in meters. Besides, $z_0$ both for the snow cover and snow-free period was calculated excluding the roof sector (from 180° to 240°) to avoid the mixing of different surface covers, even if the $z_0$ values did not change significantly taking in consideration all sectors together. The new values are now reported in the text.

The results gave $z_0 = 0.005$ m if the whole measurement period is considered. Separating the period with snow coverage from that without snow, the calculated values were $z_0 = 0.002$ m and $z_0 = 0.004$ m, respectively.

L403: rephrase. The average value of $F_{ACC}$ for the whole period was ….
Corrected in the revised version of the manuscript.

L405: Finally, the mean of $F_{CRS}$ …
Corrected in the revised version of the manuscript.

L420: overlap with
Corrected in the revised version of the manuscript.

L421: the village and shipping are all well outside the flux footprint (see comment to L357)

Actually, in the sentence questioned by the Reviewer we wanted to refer to the work of Eckhardt et al, 2013 and not directly to our measurements. We modified the sentence slightly to make it clearer.

In Ny-Ålesund, emission fluxes of natural aerosols can originate from soil erosion (in the summer) as well as snow resuspension and sublimation: besides, a contribution from the village itself and from shipping (Eckhardt et al, 2013) has been occasionally identified, but not quantified from our measurements. Finally, it is unclear whether a residual sea spray emission can lead to detectable emission fluxes of sea salt particles at ~1 km inland from the coast.

Furthermore, a new figure (1) has been inserted in the revised manuscript, where a mean footprint of the system for the whole measurement period is shown (Fig.1b) making it more understandable for the Reader. The Authors agree with the Reviewer, and indeed the overall average footprint doesn't extend to the sea fiord or the village. However, when considering the footprint calculated over 30 min, or even on a daily basis (see additional R.Fig.5), in some cases it reaches the village and the coastline. Within such cases, some emission events were registered.

Eckhardt, S., Hermansen, O., Grythe, H., Fiebig, M., Stebel, K., Cassiani, M., Baecklund, A., and Stohl, A.: The influence of cruise ship emissions on air pollution in Svalbard – a harbinger of a more polluted Arctic?, Atmos. Chem. Phys., 13, 8401–8409, https://doi.org/10.5194/acp-13-8401-2013, 2013.

[Figure]

**Figure 1.** (a) Location map of the study site: Ny-Ålesund (Svalbard, Norway). Purple and gold points indicate the Amundsen-Nobile Climate Change Tower and the Gruvebadet laboratory. Height contours (above sea level) are also presented. © Norwegian Polar Institute, www.npolar.no (accessed on 5/09/2022). In (b) the flux footprint of the EC system is represented (see Section 2.4) with fractions of the total flux originating within the respective contours.

[Figure]

R.Fig.5. A footprint example on daily average for our measurement system.

L439: simply throwing out all negative Vd's and continuing the analysis with the rest will bias the readings (unless the histogram is strongly bimodal and there are hardly any numbers around zero) and including these would bring the averages down. This is an important enough point that it should be discussed in some detail, and the potential biases evaluated. They may be part of the reason for your numbers being higher than measurements by others and the Zhang and Pleim & Ran models for much of the range (Fig. 8).

In order to deepen the understanding of deposition processes, we believe it is important to isolate the periods where the rate of deposition is dominating over emission, i.e. excluding periods where emissions are clearly dominating, resulting in a positive net flux. Every flux measured is still a net sum of the contributions from whatever processes take place in the time interval considered: we understand and agree with the Reviewer that it may not be a perfect isolation of deposition conditions. However, we believe it is by exclusively considering periods where the rate of deposition is dominant, that we can gather clear insights into the deposition process. In fact, our focus here is specially to quantify at what rate deposition occurs, when it happens, rather to draw a budget of the PM exchange at the site. So, even if two processes are occurring simultaneously, it is crucial for us to identify a consistent rate of deposition. In addition, we'd like to point out that separating upward and downward flux periods, or systematically removing periods of positive fluxes is a rather common practice in deposition studies (Emerson et al., 2020; Lavi et al., 2013; Contini et al, 2010; Wesely et al., 1985; Gallager et al., 1997; Nilsson and Rannik, 2001; Pryor et al., 2006).

In the comparison with the deposition models and observations, we know that, in some literature cases, the exchange velocity is considered (and not only the deposition velocity). Furthermore, in some cases the mean values are reported, in others the median ones. The Reviewer is right in writing that our deposition velocity values could be higher than the reported exchange velocities, but we believe this has been clarified in the manuscript (L477, for example).

Contini, D., Donateo, A., Belosi, F., Grasso, F. M., Santachiara, G., and Prodi F.: Deposition velocity of ultrafine particles measured with the Eddy-Correlation Method over the Nansen Ice Sheet (Antarctica), J. Geophys. Res., 115, D16202, https://doi.org/10.1029/2009JD013600, 2010.

Emerson, E. W., Hodshire, A. L., De Bolt, H. M., Bilsback, K. R., Pierce, J. R., McMeeking, G. R., and Farmer, D. K.: Revisiting particle dry deposition and its role in radiative effect estimates, PNAS, 117:26076–82, 2020.

Gallagher, M. W., Beswick, K. M., Duyzer, J., Westrate, H., Choularton, T. W., and Hummelshøj, P.: Measurements 835 of aerosol fluxes to speulder forest using a micrometeorological technique, 31, 359–373, https://doi.org/10.1016/S1352-2310(96)00057-X, 1997

Lavi, A., Farmer, D. K., Segre, E., Moise, T., Rotenberg, E., Jimenez, J. L., and Rudich, Y.: Fluxes of fine particles over a semi-arid pine forest: possible effects of a complex terrain, Aerosol Sci. Technol., 47, 906–915. https://doi.org/10.1080/02786826.2013.800940, 2013

Nilsson, E. D., and Rannik, U.: Turbulent aerosol fluxes over the Arctic Ocean 1. Dry deposition over sea and pack ice, J. Geophys. Res., 106(D23), 32125–32137, https://doi.org/10.1029/2000JD900605, 2001.

Pryor, S. C.: Size-resolved particle deposition velocities of sub-100nm diameter particles over a forest, Atmospheric Environment, 40, 6192–6200, https://doi.org/10.1016/j.atmosenv.2006.04.066, 2006.

Wesely, M.L. and Hicks, B.B.: A review of the current status of knowledge on dry deposition, Atmos. Environ., 34:2261–82, 2000

Table 4: did you also try doing the statistics for Vd⁻? Could be interesting and lend some support to the validity of separating out all the positives into Vd⁺ and simply ignoring negative values.

Authors agree with the Reviewer and in Table 4, Vd⁻ (emission velocity) statistics has been also reported, as follow:

| $V_d^-$ (mm s$^{-1}$) | UFP | ACC | CRS |
|---|---|---|---|
| $D_{gm}$ (µm) | 0.035 | 0.42 | 1.45 |
| 10$^{th}$ prc. | -5.73 | -3.70 | -15.80 |
| mean | -3.44 | -1.68 | -8.07 |
| std.dev | 18.68 | 4.12 | 18.84 |
| std.err | 0.41 | 0.08 | 0.27 |
| median | -0.87 | -0.77 | -5.21 |
| 90$^{th}$ prc. | -0.13 | -0.17 | -1.71 |

Vd- statistics were reported also in Table 5, as follow:

| $V_d^-$ (mm s$^{-1}$) | UFP | | | ACC | | | CRS | | |
|---|---|---|---|---|---|---|---|---|---|
| | SC | SM | SF | SC | SM | SF | SC | SM | SF |
| 10$^{th}$ prc | -6.77 | -6.60 | -3.82 | -2.48 | -4.97 | -4.67 | -12.36 | -20.21 | -19.42 |
| mean | -4.51 | -2.52 | -1.73 | -1.32 | -2.28 | -2.05 | -6.80 | -10.10 | -8.87 |
| std.dev | 24.69 | 4.65 | 2.93 | 4.54 | 4.12 | 2.34 | 14.92 | 9.85 | 9.22 |
| std.err | 0.73 | 0.24 | 0.13 | 0.12 | 0.18 | 0.11 | 0.43 | 0.42 | 0.42 |
| median | -0.89 | -0.91 | -0.80 | -0.52 | -1.13 | -1.29 | -4.24 | -7.40 | -5.97 |
| 90$^{th}$ prc | -0.13 | -0.18 | -0.12 | -0.14 | -0.21 | -0.26 | -1.57 | -2.39 | -1.75 |

Fig. 8d) snow free

Corrected in the revised version of the manuscript.

L490: Seems unlikely that water surfaces have anything to do with this since a snow surface, with all its fractal roughness elements, would be a good surface for deposition. Seems more likely that the increasing $z_0$ as the snow melts has an effect, as you reasonably state in the conclusions [L 574].

We thank the Reviewer for this remark. In the revised manuscript the sentence has been rephrased to better clarify its meaning as follow:

From the graphs in Fig. 8, $V_d^+$ (median values) increases with particle diameter, and $V_d^+$ is greater in all size bins during the melting phase with respect to the other periods. This could probably be due to the increase of roughness length $z_0$ as the snow melts (and consequent shallow ponds formation), which leads to an increased capture of the particles (in all size ranges) by interception and impaction mechanisms.

L534 and conclusions: state clearly that the polynomial only applies to the size range $0.25 - 3$ μm and does not cover UFP.

We thank the reviewer for this remark. This information, now, has been stated clearly in the revised text.

---

## Author Comment (AC2)

**Characterization of size-segregated particles turbulent flux and deposition velocity by eddy correlation method at an Arctic site. A. Donateo et al. - Responses to #RC2.**

We thank the Reviewer for her/his comments and feedback and for taking the time to help improve this manuscript. Reviewer comments are presented in blue, and sections that have been added to the text are coloured green. Author's responses are reported in black. Note: Figures in the manuscript are referred to as "Fig. X", figures included in these review responses, but not in the manuscript are referred to as "R.Fig. Y".

Donateo et al. present fluxes of particles in ultrafine, accumulation and quasi-course mode over an Arctic site. They use these data to evaluate models of size-dependent particle deposition models. This is a region that is in great need of measurements, and this manuscript offers useful data to the community.

Major Comments

I do have concerns over the placement of the flux measurements. The manuscript indicates that the measurements were on a top of a building on a mast: photos would be useful, but more importantly, we need to understand the potential for the building to impact the turbulence. We need a good diagram of the inlet setup.

A comprehensive response has been provided to a similar question by the Reviewer RC1 in this ACP open discussion (RC1: 'Comment on acp-2022-768' - https://doi.org/10.5194/acp-2022-768-RC1). The mast is 4 m tall above roof level (R.Fig.1a). The building is 5 m high (above ground level) (R.Fig.1b). The anemometer sensing volume and head of the inlet is height 60 cm and 50 cm, respectively. The total height above the ground level is 9.60 m (in the manuscript is reported as "about 10 m", now "9.6 m"). The short side of the building is 12 m, and the long side is 22 m (R.Fig.1c). The mast is located very close to roof edge (northeast-facing side). Possible flow distortions could arise for the presence of the rooftop in the wind direction sector between 180° (S) and 240° (SW) (see R.Fig.1c – yellow sector). Winds coming from this sector is uncommon in this measurement campaign (about 5% of quality assured data) (Fig.3f). Furthermore, the momentum flux for the roof wind sector (see above) did not show positive anomalies (R.Fig.2). An analysis has been performed on spectra for u, v, and w wind components from the roof sector (R.Fig.2b). From this graph can be noted that both the vertical and the horizontal wind components follow very well the trend line (black

dashed line) at -5/3 in the inertial subrange, without any evident anomaly. Thus, it doesn't seem to be any important flow distortions arising from the roof sector. For these reasons the Authors decided to take in consideration all the quality assured dataset, also including the wind sector corresponding to the roof area. In the new version of the paper, Authors decided to produce a Supplementary material file. Thus, R.Fig.1 was inserted in this Supplementary file as Fig.S1.

[Figure]

R.Fig.1. (a) Picture of the EC mast located on the roof of Gruvebadet lab and of (b) a front view of the building. (c) A sketch of the top and front view of Gruvebadet lab and EC setup.

[Figure]

R.Fig.2. (a) Momentum flux data points measured in the wind sector between 180° and 240° N. Please, note that the data points are not consecutive in time. (b) Normalized median spectra for u, v and w wind components as a function of the normalized frequency $f_n$ selected from the roof sector. Black dashed line represents the slope theoretically predicted in the inertial subrange.

In R.Fig.1a, a picture of the inlet head is reported. It was outside on the pneumatic mast, alongside the anemometer head. The air sample was transported to the instruments, inside the lab, with a silicon tube (as described in the text). At the other end of the tube, a flow splitter (R.Fig.1b) is installed, to divide the air sample between CPC and OPC. R.Fig.1 was inserted in the Supplementary material file as Fig.S2.

[Figure]

R.Fig.1. A picture of the a) inlet head with a metallic hat for precipitation repair and b) the flow splitter.

The particle losses are unclear: the authors suggest that because the D50 cut off is 5 nm, they consider their system to detect particles between 5-1000nm. That is a strange assumption: the particle losses should have substantial size dependences, meaning that some particles in the size range of a given measurement will be far more effectively sampled - and as the paper shows, the size dependence of the fluxes can be substantial even within a single mode. A figure showing the size-dependent losses, and a quantitative discussion of how those losses might introduce biases to the analysis are essential.

In R.Fig.2 we reported the inlet penetration efficiency and the relative losses for the CPC (mainly diffusional) and OPC (mainly gravitational), respectively, calculated according to Hinds (1999) and Baron and Willeke (2001). The total particle losses amount to 9% (on average) for CPC and about 2.3% (on average) for OPC. Particle losses of the measurement inlet was now described clearer in the manuscript (L121-L125). Flux is affected by particle losses at the same way as particle concentration, consequently the deposition velocity should be not affected or marginally overestimated by particle losses being calculated from the ratio of flux on concentration. R.Fig.2 was inserted in the Supplementary material file as Fig.S3.

[Figure]

R.Fig.2. Inlet penetration efficiency. a) diffusional particle losses (both for the 25 mm and 6 mm tube) and the total penetration efficiency for the CPC. b) gravitational particle losses (both for the 25 mm and 4 mm tube) and the total penetration efficiency for the OPC. Total inlet losses (in black) of the system on the right axis for each panel.

Baron, P. A. and Willeke, K.: Aerosol Measurement: Principles, Techniques, and Applications, 2nd ed., John Wiley and Sons, New York, 2001.

Hinds, W.C.: Aerosol Technology, Properties, Behaviour, and Measurement of Airborne Particles, second ed. John Wiley and Sons, New York. 1999.

Further, an important piece of QA/QC for eddy covariance fluxes is the cospectrum of the sonic anemometer data. The data shown in Figure 2 are disconcerting, as they suggest that the sensible heat flux data do not follow the expected decay. The authors state similarity to a -7/3 slope, but the plot is on a log-log curve. What actual slope is the data (i.e. -x/3?). This suggests a substantial loss of flux or disruption of the turbulence regime. You are most definitely not in the inertial subrange. I think the authors must add a substantial paragraph acknowledging these problems and considering how they would bias the data.

The Authors agree with the Reviewer. In the original figure (Fig.2a) the kinematic heat flux has been calculated on decimated time series (w and $T_s$ at 1 Hz) to obtain the same frequency of CPC and OPC (1 Hz). However, in this way heat flux did not clearly show its real trend. So, we decided to make again the Fig.2a, reporting the three cospectra each with its normalized frequency. At the same time the data binning has been modified to better represent the cospectral results. Now, in this new figure, the -7/3 trend (Kaimal and Finnigan, 1994) of the scalar cospectra is clearer (R.Fig.3a). CPC and OPC cospectra show their losses at high frequencies, as expected, due to their low time response. As clearly stated in the manuscript (L278-L286) the high frequency losses for CPC and OPC have been corrected using a parametric/in situ procedure proposed by Horst (1997). According to this frequency correction an average loss of about 30% and 21% has been calculated for CPC and OPC, respectively (L282-L283).

[Figure]

R.Fig.3. (a) Normalized median cospectra of kinematic heat flux (blue), UFP (red) and ACC-CRS particle flux (green) as a function of $f_n$. The binned median cospectra (about 1500) were computed from continuous period of 1h. Black dashed line represents the slope theoretically predicted in the inertial subrange (Kolmogorov, 1941); (b) median normalized ogives for kinematic heat flux as a function of natural frequency (Hz) for the three classes of atmospheric stability.

An analysis has been performed on spectra for u, v, and w wind components taking in consideration the whole dataset (R.Fig.3b). From this graph it can be noted that both the vertical and the horizontal wind components follow very well the trend line (black dashed line) at -5/3 in the inertial subrange,

without any evident anomaly (Kaimal and Finnigan, 1994; Stull, 1988). In Fig.2a the line with -7/3 slope is reported. The cospectra in the figure are represented on log-log scale (Schiavon et al., 2019). Actually, they are not log-cospectra.

Horst, T.W.: A simple formula for attenuation of eddy fluxes measured with first order-response scalar sensor. Boundary-Layer Meteorol. 82, 219–233, https://doi.org/10.1023/A:100022913, 1997.

Kaimal, J. C. and Finnigan, J. J.: Atmospheric Boundary Layer flows, Oxford University Press, New York, Oxford, 2nd Edn, 1994.

Schiavon, M., Tampieri, F., Bosveld, F. C., Mazzola, M., Trini Castelli, S., Viola, A. P., Yagüe, C.: The Share of the Mean Turbulent Kinetic Energy in the Near-Neutral Surface Layer for High and Low Wind Speeds, Bound-Lay. Meteor., 172:81–106. https://doi.org/10.1007/s10546-019-00435-6, 2019.

Stull, R. B.: An introduction to boundary layer meteorology, Kluwer Academic Publishers, Dordrecht, 1988.

I generally disagree with the approach of fitting the Vdep/u* versus size range to a polynomial as this is driven purely by the data, and not at all by the underlying process. I think this needs to be made exceptionally clear that this is purely a fit, not a parameterization - and that it only applies to the accumulation / quasi-coarse mode (that should be noted in the Abstract as well).

The Authors agree with the Reviewer about the approach of fitting the deposition velocity to a polynomial because it is driven by data and not by the chemical-physical processes. However, the Authors decided to include this type of analysis to have also a comparison with other works (i.e. Deventer et al, 2015). Now, in the revised text (also in the Abstract and in the Conclusion section), it has been stated clearly that the fit was applied only to ACC and CRS particles size range, excluding the UFP size bin.

Deventer, M. J., Held, A., El-Madany, T.S., Klemm, O.: Size-resolved eddy covariance fluxes of nucleation to accumulation mode aerosol particles over a coniferous forest, Agr. Forest Meteor., 214-215, 328–340, https://doi.org/10.1016/j.agrformet.2015.08.261, 2015.

Overall, this manuscript includes unique data and is well-written. I do hold concerns over the measurement site, and request that the authors provide substantial additional details and consider the implications of not measuring fluxes in the inertial subrange. Further, the authors need to be more careful over their discussion of polynomial fits. However, the data are intriguing, and warrant publication after revision in this journal.

Minor Comments

- The Introduction is well-written, but quite difficult to read as the bulk of it is in a single paragraph. I encourage the authors to consider breaking the main paragraph into about three paragraphs, each addressing the different aspects of the motivation and background.

Corrected in the revised version of the manuscript. Thanks for the suggestions.

---

## Author Response (AR2)

**Characterization of size-segregated particles turbulent flux and deposition velocity by eddy correlation method at an Arctic site**

We thank the Reviewers for their comments and feedback and for taking the time to help improve this manuscript.

**Responses to Reviewer #2.**

I commend the authors for significantly improving the manuscript by paying close attention to the reviewers' questions and recommendations. In my opinion the paper is now ready for publication, despite some outstanding issues that are difficult to fix after the fact, as outlined below. As long as there is a record (in these online comments) of some reservations, the value this paper adds to the literature on aerosol fluxes outweighs the problems.

Authors thank the Reviewer for the positive and encouraging evaluation of this work, and for giving us the chance to improve the paper. We appreciate that the Reviewer found our work interesting and relevant for the Journal and the aerosol community.

Adding photos and a schematic of the setup clarified some of the issues. It is unfortunate that there were essentially no winds from between 330° and 120°, which are the directions for which the building may have caused the least wind distortion. Judging from the new Fig. S1, any flow from between 120° and 330° would be crossing the roof to some extent, which may explain some of the positive momentum fluxes observed, particularly during strong winds. Rather than not performing a rotation for angles of attack greater than 15°, it may be safer to just exclude these data (which only account for 3% anyway).

As can be seen from Fig.3f, winds coming from the mentioned sector are uncommon (about 5% of quality assured data measured in this campaign). However, we'd like to remark that the momentum flux for the roof wind sector (identified more specifically in the range 180°-240°) did not actually show positive anomalies as shown in R.Fig.2a (https://doi.org/10.5194/acp-2022-768-AC1). Furthermore, an analysis has been performed on spectra for u, v, and w wind components from the roof sector (see R.Fig.2b, https://doi.org/10.5194/acp-2022-768-AC1): both the vertical and horizontal wind components follow very well the trend line (black dashed line) at -5/3 in the inertial subrange, without any evident anomaly. Thus, there doesn't seem to be any important flow distortion arising from the roof sector. For these reasons the Authors decided to take in consideration all the quality assured dataset, including the wind sector corresponding to the roof area. Looking at the graph of the momentum flux time series (Fig.S5) as a function of the wind direction (on colormap), a particular event can be observed (first red circle from left), with positive values of momentum flux. This positive peak corresponds to a very well-defined wind sector W (from 250° to 270°), that is, air masses coming from the glacier Brøggerbreen, SW of our measurement site, with relatively high velocity (on average 8 m s$^{-1}$). The same can be said for the second (in time) red circle event in Fig.S5.

The covariance spectra in Fig. 2 are still troublesome. Assuming that aerosols are transported by similar mechanisms as sensible heat (which we know is a stretch sometimes), why do the aerosol

covariances drop by 6 orders of magnitude from fn = 0.001 to 1, whereas heat flux covariances drop by only 2?

We believe, as specified in the description of the field setup, that the limiting factor for the frequency loss lies in the low time response of the instruments used, that are suitable for eddy covariance measurements, but still cannot cover the full turbulence spectrum. Nevertheless, the impact of such losses on our results has been minimized by applying corrections based on the approach proposed by Horst (1997)

Horst, T.W.: A simple formula for attenuation of eddy fluxes measured with first order-response scalar sensor. Boundary-Layer Meteorol. 82, 219–233, https://doi.org/10.1023/A:100022913, 1997.

The inertial subrange (where the -7/3 slope applies) should be in the same frequency range. The fact that the ogives look good is nice, but they represent normalized fractions and don't say much about the total flux which may be seriously underestimated, due to various losses. It would have been interesting to compare the aerosol to gas flux spectra, if available, which would be a better case for the assumption of similarity; perhaps a consideration for future projects.

Unfortunately, no other gaseous species were measured at the same location, to be used in comparison for better understanding the PM dynamics. However, sensible heat flux does give important information on the same turbulent exchanges involving PM, hence its extended use in eddy covariance studies. We agree that, in principle, comparing with gas fluxes could provide additional relevant information.